# Full Swap Regret and Discretized Calibration

**Maxwell Fishelson**                                       MAXFISH@MIT.EDU
*MIT*

**Robert Kleinberg**                                    RDK@CS.CORNELL.EDU
**Princewill Okorafor**                                   PCO9@CORNELL.EDU
*Cornell*

**Renato Paes Leme**                              RENATOPPL@GOOGLE.COM
**Jon Schneider**                                       JSCHNEI@GOOGLE.COM
**Yifeng Teng**                                          YIFENGT@GOOGLE.COM
*Google Research NYC*

**Editors:** Gautam Kamath and Po-Ling Loh

## Abstract

We study the problem of minimizing swap regret in structured normal-form games. Players have a very large (potentially infinite) number of pure actions, but each action has an embedding into $d$-dimensional space and payoffs are given by bilinear functions of these embeddings. We provide an efficient learning algorithm for this setting that incurs at most $\tilde{O}(T^{(d+1)/(d+3)})$ swap regret after $T$ rounds.

To achieve this, we introduce a new online learning problem we call *full swap regret minimization*. In this problem, a learner repeatedly takes a (randomized) action in a bounded convex $d$-dimensional action set $\mathcal{K}$ and then receives a loss from the adversary, with the goal of minimizing their regret with respect to the *worst-case* swap function mapping $\mathcal{K}$ to $\mathcal{K}$. For varied assumptions about the convexity and smoothness of the loss functions, we design algorithms with full swap regret bounds ranging from $O(T^{d/(d+2)})$ to $O(T^{(d+1)/(d+2)})$.

Finally, we apply these tools to the problem of online forecasting to minimize calibration error, showing that several notions of calibration can be viewed as specific instances of full swap regret. In particular, we design efficient algorithms for online forecasting that guarantee at most $O(T^{1/3})$ $\ell_2$-calibration error and $O(\max(\sqrt{\epsilon T}, T^{1/3}))$ *discretized-calibration* error (when the forecaster is restricted to predicting multiples of $\epsilon$).

**Keywords:** Swap Regret, Online Learning, Calibration

## 1. Introduction

Regret minimization is a fundamental paradigm in the theory of online learning with many applications to game theory. Perhaps most notably, it has long been known that if two players choose their actions in a repeated zero-sum game by running a learning algorithm that minimizes their *external regret*, they will over time converge to the unique Nash equilibrium (the "minimax equilibrium") of the game. This simple observation is at the core of multiple recent successes at developing superhuman-level AIs at games like Go and Poker (Silver et al., 2017; Brown and Sandholm, 2018), as well as being the main technical tool behind innovations such as boosting and GANs (Goodfellow et al., 2020; Freund and Schapire, 1996).

However, many important games in practice (such as auctions, markets, and bargaining) are not zero-sum but instead *general-sum*; depending on the outcome, both players might be better or worse

off, and there is some incentive for the players to cooperate. In such games, the theoretical guarantees of external regret minimization are markedly weaker; in particular, in general-sum games, the play of external regret minimizing algorithms only converges to the class of coarse correlated equilibria (a fairly crude relaxation of Nash equilibria), and these algorithms are also susceptible to manipulation by a strategic agent (Braverman et al., 2018; Deng et al., 2019b).

For this reason, in general-sum games, it is common to consider a more stringent benchmark known as *swap regret*. While external regret measures how much an agent could have improved their utility by switching to a single fixed action, swap regret measures how much an agent could have improved their utility by applying an arbitrary "swap function" to their sequence of past actions. Minimizing swap regret guarantees that agents converge to the sharper class of correlated equilibria, and learning algorithms that minimize swap regret are robust to the previously mentioned manipulations (Blum and Mansour, 2007; Deng et al., 2019b).

Unfortunately, swap regret is a much harder quantity to minimize than external regret. In general, a player with $n$ actions available to them each round can guarantee a worst-case external regret bound of $O(\sqrt{T \log n})$ after $T$ rounds, but only a worst-case swap regret bound of $\tilde{O}(\sqrt{nT})$. In many real-life games, the number of actions $n$ can be extremely large (e.g. all pure strategies in an extensive-form game such as poker), and so this polynomial dependence on $n$ is troubling. In recent work, Dagan et al. (2023) and Peng and Rubinstein (2023) give alternate low swap regret algorithms with a much better dependence on the number of actions, but at the cost of an exponential dependence on the amount of swap regret: to achieve $\epsilon T$ swap regret, these algorithms require $\exp(\Omega(1/\epsilon))$ rounds. Furthermore, both Dagan et al. (2023) and Peng and Rubinstein (2023) prove lower bounds showing that these rates are somewhat necessary: in the absence of any structure on the actions, any no-swap regret algorithm must incur regret that is either polynomial in the number of actions or run for a number of rounds that is exponential in the average regret per round.

## 1.1. Structured, low-dimensional games

In this paper, we consider the problem of minimizing swap regret in games where the strategy sets of the players have a low-dimensional structure, allowing us to sidestep the aforementioned lower bounds. Formally, we consider games between two players, who we will call the *Learner* (who has $n$ actions) and the *Adversary* (who has $n'$ actions). Each of these actions has a corresponding "embedding" in a lower-dimensional Euclidean space: in particular, the Learner's $n$ actions correspond to $d$-dimensional vectors $v_1, v_2, \ldots, v_n \in \mathbb{R}^d$, and the Adversary's $n'$ actions correspond to the $d$-dimensional vectors $w_1, w_2, \ldots, w_{n'} \in \mathbb{R}^d$. The payoff the Learner receives[1] when they play action $i$ and the Adversary plays action $j$ is given by $\langle v_i, w_j \rangle$. We call such games *$d$-dimensional structured games*.

Structured games encapsulate important classes of games such as Bayesian games, extensive-form games, and convex games, where the number of pure strategies is far larger (often exponentially so) than the underlying dimension of these strategies. Structured games also are a helpful abstraction of normal-form games when the two players have very different numbers of actions; any such game is a $\min(n, n')$-dimensional structured game (see Lemma 25 in Appendix D.1). Finally, structured games

---

1. In general, we consider adversarial online learning environments where our only goal is to minimize the Learner's regret, and thus where we only need to consider the Learner's utility. In results where the payoff of the Adversary is also relevant (e.g. computation of correlated equilibria), we assume the Adversary's payoff is also a bilinear function of (possibly different) embeddings of these actions into $d$-dimensional space.

capture the problem of producing *online calibrated predictions*, which will serve as a motivating example throughout this paper (and which we will introduce shortly).

In our first set of main results, we prove that it is possible to obtain swap regret bounds where the total number of rounds to achieve $\epsilon$ per-round swap regret scales as $(1/\epsilon)^{O(d)}$ and *independently* of the number of actions. In addition, the algorithms achieving these bounds are efficient, running in time polynomial in the time-horizon and dimension and again independent of the number of actions (granted access to an efficient convex decomposition oracle, that in $\mathrm{poly}(d)$ time takes an $x \in \mathrm{conv}(\{v_1, v_2, \ldots, v_n\})$ in embedding space and returns a mixed action which embeds to $x$). Specifically, we have the following theorem.

**Theorem 1** *There exists a learning algorithm for the Learner which incurs at most $\tilde{O}(T^{(d+1)/(d+3)})$ swap regret against any Adversary in any $d$-dimensional structured game. Equivalently, the Learner can guarantee $\epsilon$ per-round swap regret as long as $T = \tilde{\Omega}(1/\epsilon)^{(d+3)/2}$. This algorithm runs in per-iteration time $\mathrm{poly}(d, T)$ (assuming access to an efficient convex decomposition oracle).*

One immediate consequence of Theorem 1 is that as long as the embedding space is constant-dimensional, it is possible to design decentralized learning dynamics that converge to an $\epsilon$-correlated equilibrium in time and number of steps that is polynomial in $1/\epsilon$.

**Corollary 2** *Let $G$ be any $d$-dimensional structured game between two players. There exists a pair of learning algorithms for these two players such that if each player selects strategies according to their algorithm, the average distribution of play after $T = \tilde{\Omega}(1/\epsilon)^{(d+3)/2}$ rounds is guaranteed to be an $\epsilon$-correlated equilibrium of $G$. In addition, if both players have access to efficient convex decomposition oracles, both algorithms run in per-iteration time $\mathrm{poly}(d, T)$; in particular, it is possible to compute an $\epsilon$-correlated equilibrium in this game in time $(1/\epsilon)^{O(d)}$.*

### 1.2. Full swap regret for online learning

We prove Theorem 1 by reducing the relevant problem to another natural problem: *full swap regret minimization*. In the standard setting of online learning, a learner must, in every round $t$ (for $T$ rounds), play an action $x_t$ belonging to a $d$-dimensional bounded convex set $\mathcal{K} \subset \mathbb{R}^d$ (in fact, we will actually let the learner play a finitely-supported distribution $\mathbf{x}_t \in \Delta(\mathcal{K})$ over such actions). Then, a loss function $\ell_t : \mathcal{K} \to \mathbb{R}$ is revealed by the adversary and the learner incurs loss[2] $\ell_t(\mathbf{x}_t) := \mathbb{E}_{s \sim \mathbf{x}_t}[\ell_t(s)]$. The standard objective in online learning is to minimize the *external regret*, defined as

$$\mathsf{ExtReg} := \max_{x^* \in \mathcal{K}} \sum_{t=1}^{T} [\ell_t(\mathbf{x}_t) - \ell_t(x^*)].$$

We consider a swap regret variant of this objective, where instead of competing against the best fixed action $x^* \in \mathcal{K}$ in hindsight, we compete against the best arbitrary swap function $\phi : \mathcal{K} \to \mathcal{K}$ in hindsight.

---

2. Traditionally, this loss function is also assumed to be convex. In the majority of our applications this will be the case, but we will also consider some settings with weaker constraints on the $\ell_t$ (e.g., Lipschitz or concave $\ell_t$).

$$\mathsf{FullSwapReg} := \max_{\phi:\mathcal{K}\to\mathcal{K}} \sum_{t=1}^{T} \left[ \ell_t(\mathbf{x}_t) - \ell_t(\phi(\mathbf{x}_t)) \right].$$

Here $\phi(\mathbf{x}_t) \in \Delta(\mathcal{K})$ is the distribution of $\phi(s)$ where $s \sim \mathbf{x}_t$. We call this quantity ***full** swap regret* to emphasize a distinction from other notions of swap regret where the class of transformations $\phi$ is restricted in some way (e.g., linear swap regret, where $\phi$ must also be a linear transformation).

We provide a family of algorithms to minimize full swap regret over convex action sets. The optimal rates achievable depend on the qualitative properties of the losses faced by the learner (in particular, whether they are strongly convex, smooth, both, or neither).

**Theorem 3 (Informal version of Theorem 15)** *Let $\mathcal{K} \subseteq \mathbb{R}^d$ be a convex set of diameter 1. Let $\mathcal{L} \subseteq \{\ell : \mathcal{K} \to \mathbb{R}\}$ be a family of $O(1)$-Lipschitz loss functions. We attain the following* $\mathsf{FullSwapReg}$ *guarantees in terms of the following constraints on $\mathcal{L}$.*

| *For all $\ell \in \mathcal{L}$* | $\mathsf{FullSwapReg}$ *rate* |
|---|---|
| $\ell$: *no assumption* | $\tilde{O}\left(T^{\frac{d+1}{d+2}}\right)$ |
| $\ell$: *linear (or more generally, concave)* | $\tilde{O}\left(T^{\frac{d+1}{d+3}}\right)$ |
| $\ell$: $O(1)$-*smooth* | $\tilde{O}\left(T^{\frac{d+2}{d+4}}\right)$ |
| $\ell$: $\Omega(1)$-*strongly-convex* | $\tilde{O}\left(T^{\frac{d}{d+1}}\right)$ |
| $\ell$: $\Omega(1)$-*strongly-convex and $O(1)$-smooth* | $\tilde{O}\left(T^{\frac{d}{d+2}}\right)$ |

Theorem 3 (in particular, the setting with linear losses) can be immediately applied to recover Theorem 1 on minimizing swap regret in structured games. Indeed, instead of thinking of the Learner as playing a mixed strategy $\alpha_t \in \Delta_n$ supported on their $n$ pure actions, it suffices to look at the projection $x_t$ of their strategy onto their embedding space $\mathcal{K} = \mathrm{conv}(\{v_1, v_2, \ldots, v_n\}) \subseteq \mathbb{R}^d$. By the structure of the game, they face *linear losses* in this embedding space, and it is therefore possible to upper bound the swap regret the Learner incurs in the original game by the full swap regret they incur in this projected problem, which by Theorem 1 can be guaranteed to be at most $\tilde{O}(T^{(d+1)/(d+3)})$.

**Online calibration and full swap regret** In cases where the losses in our game have additional structure, it is possible to apply the other guarantees in the statement of Theorem 3 to obtain even stronger regret bounds. One important application where this is the case is the problem of producing online calibrated forecasts.

In the problem of online calibration, the Learner is a forecaster who, every round, must make a (randomized) prediction $\mathbf{x}_t \in \Delta([0, 1])$ for the probability of some binary event (e.g., whether it will rain that day or not). Then nature (the Adversary) either realizes this event (sets $b_t = 1$) or does not (sets $b_t = 0$). The quality of the learner's forecasts is evaluated via their calibration error, which essentially asks "in the rounds where the forecaster predicted 20% probability of rain, did it rain 20% of the time?". Mathematically, we define the $\ell_2$-*calibration error* as:

$$\mathsf{Cal}(\mathbf{x}_{1:T}, \mathbf{b}_{1:T}) = \sum_{p \in [0,1]} \left( \sum_t \mathbf{x}_t[p] \right) \cdot \left( p - \frac{\sum_t b_t \mathbf{x}_t[p]}{\sum_t \mathbf{x}_t[p]} \right)^2,$$

where $\mathbf{x}_t[p]$ is the probability that the forecaster predicts probability $p \in [0, 1]$ in round $t$. It can be shown (see Lemma 26 in Appendix D.2) that this calibration error $\mathsf{Cal}$ is exactly equal to the Learner's swap regret in the game where they receive utility $u(x, b) = -x^2 + b(2x - 1)$ when the Learner plays the pure strategy $x \in [0, 1]$ and the Adversary plays the pure strategy $b \in \{0, 1\}$. In particular this quantity can be interpreted as the Learner's swap regret in a two-dimensional structured game where the pure strategy $x$ corresponds to the embedding $v_x = (2x - 1, -x^2)$, and the pure strategy $b$ corresponds to the embedding $w_b = (b, 1)$.

Naively applying the bounds in Theorem 1 to this game gives us an algorithm with a calibration error of $\tilde{O}(T^{3/5})$. But since the losses in calibration are quadratic functions of the single-dimensional parameter $x$, we can directly apply the strongly convex and smooth case of Theorem 3 for $d = 1$ to instead obtain an online forecaster with $\tilde{O}(T^{1/3})$ $\ell_2$-calibration error.

**Theorem 4 ($\ell_2$-Calibration)** *There is a forecasting algorithm with $\tilde{O}(T^{1/3})$ $\ell_2$-calibration error.*

Note that this is also asymptotically better than both the bound on $\ell_2$-calibration inherited from the best bounds for $\ell_1$-calibration (Dagan et al., 2024), and the $O(\sqrt{T})$ bound obtained by naively applying Blum-Mansour to an $\epsilon$-net of predictions.

**Algorithmic techniques** All of the algorithms in Theorem 3 can be seen as extensions of the classical swap-regret minimization algorithm of Blum and Mansour (2007) for the case of discrete actions. In that algorithm, we maintain an instance of an external regret minimization algorithm for each action. In each round, each action's sub-algorithm outputs a probability distribution over the set of all actions, and taken together, these distributions can be viewed as a Markov chain over the actions. The learner then samples an action from the stationary distribution of this Markov chain, which allows us to decompose the swap regret into the sum of external regrets of each of the sub-algorithms. With $n$ discrete actions, this algorithm incurs at most $O(\sqrt{nT})$ swap regret.

The immediate difficulty of applying this idea to convex action sets is that the set of actions is infinite, so the previously mentioned regret bound is vacuous. Our solution to this problem has two main components. The first (rather natural) idea is to discretize the set of actions, apply the Blum-Mansour algorithm over this discretization, and bound the discretization error. If we choose the discretization carefully (by performing an appropriate polytope approximation of the convex set $\mathcal{K}$), this idea alone is enough to recover the first three rows of guarantees in Theorem 3.

However, this black box application of Blum-Mansour is inherently lossy – by treating every point in the discretization as an individual action, we lose the information that some pairs of actions correspond to very nearby points whereas others correspond to very dissimilar points. Our second main technical insight is to modify the Blum-Mansour algorithm to take advantage of this information by adjusting the behavior of the exteral regret minimization sub-algorithms. Indeed, we begin with a discretization $K^\epsilon$ of the action set $\mathcal{K}$, and we instantiate an external regret algorithm for each point in $K^\epsilon$. However, instead of running a generic regret minimization algorithm over the set of distributions $\Delta(K^\epsilon)$, we modify each sub-algorithm to be a combination of an online convex optimization algorithm over $\mathcal{K}$ (e.g. some variant of gradient descent) and an appropriate rounding procedure mapping $\mathcal{K}$ to $\Delta(K^\epsilon)$. This allows the subroutines to attain improved rates when the losses are strongly-convex (where we can obtain external regret bounds of $O(\log T)$ instead of $O(\sqrt{T})$), and allows us to achieve the remaining guarantees in Theorem 3.

### 1.3. Discretized calibration: interpolating between linear and strongly convex losses

Carefully examining the regret guarantees in Theorem 3 reveals a mathematically counter-intuitive phenomenon. Let's consider the case $d = 1$. If we focus on one extreme of the space of possible loss functions – the case of linear losses – it is possible to obtain swap regret bounds that scale as $\tilde{O}(\sqrt{T})$ (row 2). On the other hand, if we look at another extreme – strongly convex loss functions – we again attain $\tilde{O}(\sqrt{T})$ regret (row 4). But if we interpolate between these two extremes and look at sequences of loss functions that are merely guaranteed to be convex, the best regret bound that applies is simply the general regret bound of $\tilde{O}(T^{2/3})$ (row 1). This is worse than the regret bound at either of the two extremes! This raises the following question:

**Question 1** *For $d = 1$, is there an algorithm which guarantees* $\mathsf{FullSwapReg} = \tilde{O}\left(\sqrt{T}\right)$ *against any sequence of $O(1)$-Lipschitz convex loss functions $\ell_t$?*

We do not resolve Question 1 in this paper, but we provide some evidence that the answer is positive by examining some natural instances of this question where this phenomenon already occurs. One especially striking example is a natural variant of the online calibration problem that we call *discretized calibration*.

The setting of discretized calibration is almost identical to the online calibration setting described above, with the main difference being that the learner is required to make predictions that are multiples of a fixed $\epsilon$ (e.g., 5%)? This is practically motivated by the fact that in many typical forecasting settings, forecasts are often "binned" into such discrete multiples; for example, a newspaper may be reluctant to publish a weather forecast of a 13.14159% probability of rain and instead may prefer to report that there is a 15% probability of rain instead. Formally, the forecaster makes predictions $\mathbf{x}_t \in \Delta([0,1] \cap \epsilon\mathbb{Z})$ (note that this action set is a $\lceil 1/\epsilon \rceil$-dimensional simplex). We modify the calibration loss to $\mathsf{Cal}_\epsilon$ by replacing the $(p - \bar{b}(p))^2$ term with $(p - [\bar{b}(p)]_\epsilon)^2$, where $[b]_\epsilon \in [0,1] \cap \epsilon\mathbb{Z}$ rounds $b \in [0,1]$ to the nearest multiple of $\epsilon$. In other words, we only compete against predictors who also are forced to discretize their forecasts (without this constraint, we would be forced to incur $\Omega(\epsilon T)$ calibration error simply due to this rounding error).

If the learner is required to make predictions that are multiples of $\epsilon$, there are two natural approaches. The first is to run our algorithm in Theorem 4 for general (non-discretized) calibration, and round each prediction to the nearest multiple of $\epsilon$. This leads to a discretized calibration error of $\tilde{O}(T^{1/3} + \epsilon^2 T)$. A second approach is to simply treat this as a regular discrete swap regret problem over an action set with $1/\epsilon$ actions and directly run the algorithm of Blum and Mansour (2007). This leads to a discretized calibration error of $O(\sqrt{T/\epsilon})$. The calibration errors of these two approaches are depicted in Figure 1. The best of those natural approaches cannot guarantee regret below $O(\sqrt{T})$ for $\epsilon \in (T^{-1/4}, 1)$. In the following theorem we significantly improve this bound.

**Theorem 5 (Discretized $\ell_2$-Calibration)** *There is an online algorithm for $\epsilon$-discretized calibration which incurs calibration error at most $\tilde{O}(\max(T^{1/3}, \sqrt{\epsilon T}))$.*

Our algorithm for discretized calibration is a special case of a $\mathsf{SwapReg}$ algorithm for when the action set is a discretization of a one-dimensional convex set.

**Theorem 6 (Informal version of Theorem 21)** *Let $K \subseteq [0,1]$ be a discrete bounded set such that for each $x \in [0,1]$ there exists a $y \in K$ such $|x - y| \leq \epsilon$. If the losses are Lipschitz and $\alpha$-strongly*

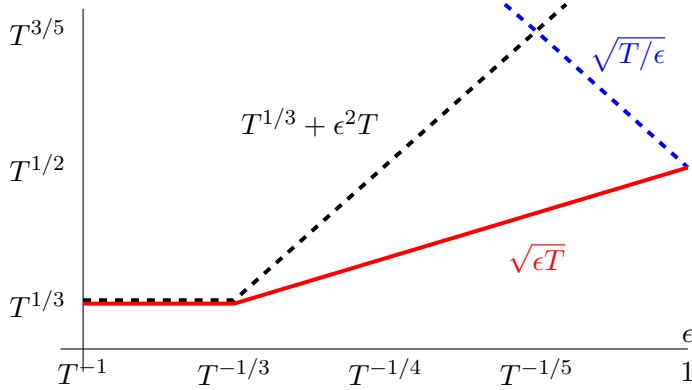

Figure 1: The $x$-axis shows the discretization parameter $\epsilon$ and the $y$-axis the calibration loss Cal for three different algorithms: dashed blue (regular swap regret on $1/\epsilon$ actions), dashed black (algorithm in Theorem 4 + rounding) and red (algorithm in Theorem 5).

*convex, then there exists an algorithm that incurs at most $O(\sqrt{\epsilon T} + \epsilon^{-1}\log T)$ swap regret (against the best map $\phi : K \to K$).*

The main idea behind the proof of Theorem 6 is to treat the problem as a full swap regret problem by replacing each loss function by its *piecewise linearization*: the continuation of the original loss function $\ell$ from the discrete domain $K$ to the extended domain $[0, 1]$ by taking the lower convex hull of the set of points $\{(x, \ell(x))\}_{x \in K}$. While the original losses are strongly convex, these piecewise linearizations are not, and so we cannot immediately apply the results of Theorem 3. However, they are what we call $(\alpha, \epsilon)$-*nearly-strongly-convex*, i.e. they satisfy the strong-convexity property for points that are at least $\epsilon$-apart. We develop a new external regret algorithm for nearly-strongly convex functions (that may be of independent interest); using this algorithm as the sub-routine in Blum-Mansour allows us to recover Theorem 6.

### 1.4. Related work

**Swap regret**  There is a large line of recent work on designing low-swap regret algorithms and understanding their game-theoretic guarantees (Braverman et al., 2018; Deng et al., 2019a,b; Camara et al., 2020; Mansour et al., 2022; Cai et al., 2023; Brown et al., 2024; Haghtalab et al., 2024). Almost all of these results are constrained to the discrete setting for low dimension. Very recently Dagan et al. (2023) and Peng and Rubinstein (2023) presented algorithms that achieve full swap regret of $O(T/\log T)$, independent of $d$. This work improves on these results in the regime $d = o(\log(T)/\log\log(T))$. Roth and Shi (2024) and Hu and Wu (2024) study the problem of designing forecasts such that any downstream agent incurs low swap regret; the problem faced by a single downstream agent (with potentially many actions but where the payoff only depends on a low-dimensional outcome) can be interpreted as a swap-regret minimization problem in a structured game.

$\phi$**-regret**  Gordon et al. (Gordon et al., 2008) introduced a generalization of swap regret called $\phi$-regret, where instead of competing only with standard swaps you compete with all transformation

functions $\phi : \mathcal{K} \to \mathcal{K}$ belonging to a set $\Phi$. Our full swap regret corresponds to the extreme case of this where $\Phi$ contains every such function. Mansour et al. (2022), Farina and Pipis (2024), and Daskalakis et al. (2024) study a variant of $\phi$-regret called LinearSwapReg that competes only against linear maps $\phi$. This is a much weaker notion than FullSwapReg and does not suffice for obtaining bounds on calibration error. Other variants of $\phi$-regret minimization have found applications in designing learning algorithms for extensive-form games (Celli et al., 2021; Bai et al., 2022; Zhang et al., 2024).

**Calibration** There is a well-established connection in the literature between calibration error and swap-regret (Cesa-Bianchi and Lugosi, 2006) (in fact the very first low swap-regret algorithms worked via responding to calibrated loss estimates, Foster and Vohra (1997)). Designing online forecasters with good calibration guarantees is a problem of major interest (Brier, 1950; Murphy, 1972, 1973; Foster and Vohra, 1998; Kleinberg et al., 2023; Gopalan et al., 2022), with the optimal online bounds for $\ell_1$ calibration a major open problem (Qiao and Valiant, 2021). Recent work of Dagan et al. (2024) improve upon the bounds for $\ell_1$ calibration from Qiao and Valiant (2021). However, our work focuses on $\ell_2$ calibration, as the $\ell_1$ calibration loss cannot be written as the swap regret in some game.

**Structured games** Our definition of $d$-dimensional structured games is similar in many ways to the definition of convex games (also appearing under the names "polyhedral games" or "polytope games" in the literature), where both players pick actions in a convex set and obtain utility given by a bilinear function of both players' actions (Gordon et al., 2008; Chakrabarti et al., 2024; Mansour et al., 2022). We use the term "structured game" throughout this paper to emphasize the fact that when computing the swap regret, the swap functions act on the pure strategies in the associated normal-form game, not on the embeddings (i.e., we think of such a game as a large normal-form game with additional structure).

## 2. Preliminaries

### 2.1. Structured Games and Swap Regret Minimization

A normal-form game with $n$ actions for the first player (the Learner) and $n'$ actions for the second player (the Adversary) is a $d$-dimensional structured game if:

- The utility $u_L(i, j)$ the Learner receives when the Learner plays pure action $i \in [n]$ and the Adversary plays pure action $j \in [n']$ can be expressed in the form $\langle v_i, w_j \rangle$ for some $v_1, v_2, \ldots, v_n \in \mathbb{R}^d$ and $w_1, w_2, \ldots, w_{n'} \in \mathbb{R}^d$.

- The utility $u_A(i, j)$ the Adversary receives when the Learner plays pure action $i \in [n]$ and the Adversary plays pure action $j \in [n']$ can be expressed in the form $\langle v'_i, w'_j \rangle$ for some (potentially different) $v'_1, v'_2, \ldots, v'_n \in \mathbb{R}^d$ and $w'_1, w'_2, \ldots, w'_{n'} \in \mathbb{R}^d$.

In general, most of our results pertaining to structured games focus on minimizing the regret of only the Learner, and so only the structure of the Learner's payoff $u_L$ is relevant (the only result where the utility $u_A$ of the Adversary is also relevant is Corollary 2, on computation of correlated equilibria).

We will write $\mathcal{K} = \text{conv}(\{v_1, v_2, \ldots, v_n\})$ be the convex hull of the embeddings of the Learner's actions, and let $\mathcal{K}' = \text{conv}(\{w_1, w_2, \ldots, w_{n'}\})$ be the convex hull of the embeddings of the Adversary's actions. For any mixed strategy $\mathbf{p} \in \Delta_n$, we will let $\pi(\mathbf{p}) \in \mathcal{K}$ be the corresponding embedding provided by $\pi(\mathbf{p}) = \sum_{i=1}^{n} p_i v_i$; note that this is a surjective map from $\Delta_n$ to $\mathcal{K}$. Likewise, for any $\mathbf{q} \in \Delta_{n'}$ we will let $\pi'(\mathbf{q}) = \sum_{i=1}^{n'} q_i w_i \in \mathcal{K}'$. We will further assume that all embeddings have bounded $\ell_2$-norm; that is, $\|\mathbf{x}\| \leq 1$ for any $\mathbf{x} \in \mathcal{K}$ and $\|\mathbf{y}\| \leq 1$ for any $\mathbf{y} \in \mathcal{K}'$.

To obtain efficient learning algorithms in structured games from efficient full swap regret minimization algorithms, we will also assume that the Learner has access to an efficient *convex decomposition oracle*, which takes any $x \in \mathcal{K}$ and returns the value of $\pi^{-1}(x) \in \Delta_n$ (for some arbitrary but fixed inverse map $\pi^{-1}(x)$) in time $\text{poly}(d)$. Note that such oracles generally exist for computationally "nice" embeddings. For example, in any case where the Learner possesses a membership oracle for the set $\mathcal{K}$, the Learner can implement such a convex decomposition oracle by following the standard construction in the proof of Caratheodory's theorem (see e.g. Theorem 6.5.11 in Grötschel et al. (2012)). Even when an efficient oracle does not exist, these operations are implementable in time $O(n, n')$, and so in the worst-case this only adds a $\text{poly}(n, n')$ factor to the run-time.

**Regret minimization** We are concerned in designing learning algorithms for the Learner in cases where a structured game is played repeatedly for $T$ rounds. A learning algorithm $\mathcal{A}$ for the Learner is a collection of functions $\mathcal{A}_t$ (one for every $t \in [T]$) that map the first $t - 1$ mixed actions of the Adversary $\mathbf{q}_1, \mathbf{q}_2, \ldots, \mathbf{q}_{t-1} \in \Delta_{n'}$ to the Learner's mixed action $\mathbf{p}_t \in \Delta_n$ at round $t$. In repeated structured games, we will assume that after each round, both players receive not only the action ($\mathbf{p}$ or $\mathbf{q}$) played by their opponent, but also the corresponding embedding ($\pi(\mathbf{p})$ or $\pi'(\mathbf{q})$) of this action[3].

We evaluate the performance of a learning algorithm $\mathcal{A}$ in a repeated game via various worst-case counterfactual regret notions. The main regret notion of interest in this paper is the Learner's swap regret. The swap regret of the learner in a given transcript of the game (expressed via sequences $\mathbf{p}_1, \mathbf{p}_2, \ldots, \mathbf{p}_T$ and $\mathbf{q}_1, \mathbf{q}_2, \ldots, \mathbf{q}_T$ of Learner and Adversary actions) is given by:

$$\mathsf{SwapReg}(\mathbf{p}_{1:T}, \mathbf{q}_{1:T}) = \max_{\phi:[n] \to [n]} \sum_{t=1}^{T} \left( u_L(\phi(\mathbf{p}_t), \mathbf{q}_t) - u_L(\mathbf{p}_t, \mathbf{q}_t) \right),$$

where the swap function $\phi : [n] \to [n]$ is linearly extended to act on $\mathbf{p}_t \in \Delta_n$.

### 2.2. Online Convex Optimization

**Convexity** For a convex set $\mathcal{K} \subseteq \mathbb{R}^d$, a loss function $\ell : \mathcal{K} \to \mathbb{R}$ is convex if for all $x \in \mathcal{K}$ there exists a subgradient $\nabla \ell(x)$ such that $\ell(y) - \ell(x) - \nabla \ell(x)^\top (y - x) \geq 0$ for all $y \in \mathcal{K}$. Whenever we can strengthen it to a quadratic lower bound, we call the function $\alpha$-strongly-convex: $\ell(y) - \ell(x) - \nabla \ell(x)^\top (y - x) \geq \frac{\alpha}{2} \|y - x\|^2$ for all $x, y \in \mathcal{K}$. If the inequality holds in the opposite direction we say that the loss function is $\beta$-smooth: $\ell(y) - \ell(x) - \nabla \ell(x)^\top (y - x) \leq \frac{\beta}{2} \|y - x\|^2$.

**(Scaled) External Regret Minimization** One important primitive in our algorithms for full swap regret minimization is a variant of the standard problem of minimizing external regret in online

---

3. This assumption is made for convenience of expressing time complexity bounds in situations where $n$ or $n'$ is much bigger than $d$. Without this assumption, the player might have to spend $O(nd)$ or $O(n'd)$ time simply reading the other player's action.

convex optimization where the regret in different rounds is scaled by an adversarially provided value. In the scaled version of online learning over a convex set $\mathcal{K}$ for a family of loss functions $\mathcal{L}$, the learner chooses an action[4] $x_t \in \mathcal{K}$ and then an adversary reveals a loss $\ell_t \in \mathcal{L}$ (as usual) and a scale $g_t \in [0, 1]$. The external regret is scaled in each step: $\mathsf{ExtReg} := \max_{x^* \in \mathcal{K}} \sum_{t=1}^{T} g_t(\ell_t(x_t) - \ell_t(x^*))$ and we expect the regret bound to be given as a function of $G_T = \sum_t g_t$ instead of $T$.

The standard algorithm for this type of problem is Online Gradient Descent (OGD), which starts at an arbitrary point $x_1 \in \mathcal{K}$ and updates the estimate according to $x_{t+1} \leftarrow \Pi_{\mathcal{K}}(x_t - \eta_t g_t \nabla \ell_t(x_t))$ where $\Pi_{\mathcal{K}}$ is the $\ell_2$-projection into the convex set $\mathcal{K}$ and $\eta_t$ is a learning rate to be specified in each variation. The following bounds can be obtained from adapting the standard algorithms to the scaled setting (see Hazan (2022) for example). We include a full proof for strongly-convex losses in the appendix for completeness:

- for convex and $L$-Lipschitz losses, OGD has $\mathsf{ExtReg} \leq L\sqrt{2G_T}$

- for $\alpha$-strongly-convex and $L$-Lipschitz losses, OGD has $\mathsf{ExtReg} \leq \frac{L^2}{2\alpha}(\log(G_T + 1) + 1)$ (Lemma 24)

### 2.3. Convex Geometry, Nets, and Triangulations

In order to get low swap regret, it will be important that our learning algorithms play actions supported over a (relatively small) discretization of the full convex set. In this section we review some preliminaries from computational geometry that will be helpful for establishing these algorithms later.

**Definition 7 ($\epsilon$-net)** *For a convex set $\mathcal{K} \subseteq \mathbb{R}^d$, define an $\epsilon$-net $K^\epsilon$ of $\mathcal{K}$ to be a finite subset of $\mathcal{K}$ satisfying, for all $x \in \mathcal{K}$, there exists a point $K^\epsilon(x) \in K^\epsilon$ such $\|x - K^\epsilon(x)\| \leq \epsilon$.*

For any bounded, constant diameter, $d$-dimensional convex set $K$, we can find an $\epsilon$-net of $K$ with at most $O(\epsilon^{-d})$ points.

**Lemma 8 (Chapter 4.2 of Vershynin (2018))** *For all convex sets $\mathcal{K} \subseteq \mathbb{R}^d$ contained in the unit ball, there exists an $O(\epsilon)$-net of $\mathcal{K}$ with $|K^\epsilon| = O(\epsilon^{-d})$.*

For some of our algorithms, we will want a discretization of our set with a stronger guarantee than just being an $\epsilon$-net; we also want the ability to decompose any point in the original set as a convex combination of nearby points. This motivates the following definition of an $\epsilon$-triangulation.

**Definition 9 ($\epsilon$-triangulation)** *For a convex set $\mathcal{K} \subseteq \mathbb{R}^d$, define an $\epsilon$-triangulation $K^\epsilon$ of $\mathcal{K}$ to be a finite subset of $\mathcal{K}$ satisfying*

- *Every point in $\mathcal{K}$ is within distance $\epsilon^2$ of some point in $\mathrm{conv}(K^\epsilon)$ (i.e., $\sup_{x \in \mathcal{K}} \|x - \Pi_{\mathrm{conv}(K^\epsilon)}(x)\| \leq \epsilon^2$).*

- *Every point in $\mathrm{conv}(K^\epsilon)$ is contained inside a simplex with vertices in $K^\epsilon$ and diameter at most $2\epsilon$. That is, for all $x \in \mathrm{conv}(K^\epsilon)$, there exists a subset $K^\epsilon(x) \subseteq K^\epsilon$ such that*

---

4. Recall that in the earlier definition of online learning, we allowed the learner to choose a distribution over actions $\mathbf{x}_t \in \Delta(\mathcal{K})$. This is essential for swap regret but unnecessary for external regret, since playing $\mathbb{E}[\mathbf{x}_t] \in \mathcal{K}$ always results in at most the same external regret as playing the randomized strategy $\mathbf{x}_t$.

- $|K^\epsilon(x)| \leq d + 1$
- $x \in \mathrm{conv}(K^\epsilon(x))$
- *For all $s \in K^\epsilon(x)$, $\|x - s\| \leq \epsilon$*

*where* $\mathrm{conv}(S)$ *refers to the convex hull of $S$.*

**Lemma 10** *For any convex set $\mathcal{K} \subseteq \mathbb{R}^d$ contained within the unit ball, there exists an $O(\epsilon)$-triangulation $K^\epsilon$ of $\mathcal{K}$ with $|K^\epsilon| = O\left(\sqrt{d}\left(\frac{1}{\epsilon}\right)^d\right)$.*

This follows from a result of Bronshteyn and Ivanov (1975) with some modifications. We provide a full proof in Appendix A.

Lastly, we introduce the notation $\mathrm{dist}(x, \mathcal{P})$ to refer to the Euclidean, projective distance of a vector $x \in \mathbb{R}^d$ to a convex set $\mathcal{P} \subseteq \mathbb{R}^d$.

## 3. From Structured Games to Full Swap Regret

In this section, we will discuss how to formally reduce the problem of minimizing swap regret in a structured game to the problem of minimizing full swap regret in a related online linear optimization instance (thus establishing Theorem 1). We begin by showing that we can bound swap regret in a $d$-dimensional structured game by the full swap regret of a learning algorithm facing linear losses.

**Lemma 11** *Assume there is an learning algorithm $\mathcal{A}$ that guarantees* $\mathsf{FullSwapReg} \leq R(d, T)$ *against any sequence of $T$ $d$-dimensional $O(1)$-Lipschitz linear losses with domain $\mathcal{K}$. Then, for any $d$-dimensional structured game $G$, there exists a learning algorithm $\mathcal{A}'$ for the Learner which incurs at most $R(d, T)$ swap regret after repeatedly playing $G$ for $T$ rounds. Moreover, if the algorithm $\mathcal{A}$ has a per-round time complexity of $\tau(d, T)$ and the Learner has access to an efficient convex decomposition oracle, then the learning algorithm $\mathcal{A}'$ has a per-round time complexity of* $\mathrm{poly}(d)\tau(d, T)$.

**Proof** Let $v_1, v_2, \ldots, v_n \in \mathbb{R}^d$ be the $d$-dimensional embeddings of the $n$ pure actions of the Learner, and let $\mathcal{K} = \mathrm{conv}(\{v_1, v_2, \ldots, v_n\})$. Recall that every mixed action $\mathbf{p} \in \Delta_n$ projects to some point $\pi(\mathbf{p}) \in \mathcal{K}$ via $\pi(\mathbf{p}) = \sum_{i=1}^n p_i v_i$ and that this projection map is surjective. Fix any specific inverse map $\pi^{-1}(\mathbf{x})$ (i.e., a specific way of taking an embedding in $\mathcal{K}$ and constructing a mixed strategy that corresponds to this embedding).

Consider now an algorithm $\mathcal{A}$ for online linear optimization over the set $\mathcal{K}$. This algorithm produces a sequence of randomized actions $\mathbf{x}_t \in \Delta(\mathcal{K})$ in response to a sequence of linear losses $\ell_t \in \mathcal{L}$. We will use it to construct an algorithm $\mathcal{A}'$ for learning in the repeated structured game $G$ that works as follows:

1. Receive a randomized action $\mathbf{x}_t \in \Delta(\mathcal{K})$ from $\mathcal{A}$.

2. Play the action $\mathbf{p}_t = \mathbb{E}_{x_t \sim \mathbf{x}_t}[\pi^{-1}(x_t)] \in \Delta_n$ in the repeated game.

3. Receive the action $\mathbf{q}_t \in \Delta_{n'}$ played by the adversary in this round, and its embedding $y_t = \pi'(\mathbf{q}_t) \in \mathbb{R}^d$.

4. Define the loss $\ell_t(x) : \mathcal{K} \to \mathbb{R}$ via $\ell_t(x) = -\langle x, y_t \rangle$. Pass this loss to $\mathcal{A}$.

We want to show that the swap regret incurred by $\mathcal{A}'$ is at most the full swap regret incurred by $\mathcal{A}$. To see this, note that for any swap function $\phi : [n] \to [n]$, the utility $u_L(\phi(\mathbf{p}_t), \mathbf{q}_t)$ can be written as

$$u_L(\phi(\mathbf{p}_t), \mathbf{q}_t) = -\ell_t(\mathbb{E}_{x_t \sim \mathbf{x}_t}[\pi(\phi(\pi^{-1}(x_t)))]) = -\ell_t(\tilde{\phi}(\mathbf{x}_t)),$$

for the function $\tilde{\phi}(x) : \mathcal{K} \to \mathcal{K}$ given by $\tilde{\phi}(x) = \pi(\phi(\pi^{-1}(x)))$. It follows that

$$\mathsf{SwapReg}(\mathbf{p}_{1:T}, \mathbf{q}_{1:T}) = \max_{\phi:[n]\to[n]} \sum_{t=1}^{T} \left( u_L(\phi(\mathbf{p}_t), \mathbf{q}_t) - u_L(\mathbf{p}_t, \mathbf{q}_t) \right)$$

$$\leq \max_{\tilde{\phi}:\mathcal{K}\to\mathcal{K}} \sum_{t=1}^{T} \left( \ell_t(\mathbf{x}_t) - \ell_t(\tilde{\phi}(\mathbf{x}_t)) \right) = \mathsf{FullSwapReg}(\mathbf{x}_{1:T}, \ell_{1:T}).$$

Finally, note that with access to a best response oracle, computing $\pi^{-1}(x_t)$ takes at most $\mathrm{poly}(d)$ time, and so computing $\mathbf{p}_t$ takes at worst $\mathrm{poly}(d)\tau(d, T)$ time (the per-iteration time complexity of $\mathcal{A}$ cannot be better than the sparsity of $\mathbf{x}$). Overall, $\mathcal{A}'$ therefore has time complexity at most $\mathrm{poly}(d)\tau(d, T)$. ∎

Lemma 11 together with our results on full swap regret minimization (to be established in Section 5) directly implies Theorem 1 in the introduction (and in turn, Corollary 2).

**Proof** [Proof of Theorem 1] Note that by the third row of Table 5, there exists a learning algorithm $\mathcal{A}$ that guarantees $\mathsf{FullSwapReg} \leq \tilde{O}(T^{(d+1)/(d+3)})$ running in per-round time $\mathrm{poly}(d, T)$. By Lemma 11, this implies an algorithm for swap-regret minimization in $d$-dimensional structured games incurring swap-regret at most $\tilde{O}(T^{(d+1)/(d+3)})$ and running in polynomial time, as desired. ∎

## 4. Algorithmic Template: Blum-Mansour for Convex Action Spaces

All of our algorithms for full-swap-regret minimization rely on the core algorithmic idea of Blum and Mansour (2007). That idea is to create an instance of some external-regret minimizing algorithm for each action in the decision space, and at each round, output the stationary distribution of a Markov chain induced by the recommendations of these algorithms. Unmodified, this algorithm cannot attain full-swap-regret over a convex set $\mathcal{K}$, as it would require an external-regret minimizing instance for each of the infinitely many points in the set. The key idea behind all of our full-swap-regret minimizing algorithms is to create $K^\epsilon$: a discretization of the set $\mathcal{K}$.

Instead of creating an external-regret-minimizing instance for every point in $\mathcal{K}$, we only create instances for each of the finitely many points in $K^\epsilon$. For non-convex losses, our approach simply entails running Blum Mansour over this discretization (Algorithm 2 in appendix B). For strongly convex and nearly-strongly convex losses (the cases of greatest interest for calibration), we obtain significant improvement using a novel technique (Algorithm 1 in appendix B). Namely, at each time step, each external-regret-minimizing instance will produce a recommendation in $\mathcal{K}$, rather than $\Delta(K^\epsilon)$. We would like the recommendations to be distributions $\Delta(K^\epsilon)$ in order to induce a Markov chain over $K^\epsilon$. But, rather than having the external regret subroutines produce such recommendations directly, we have them recommend in $\mathcal{K}$ and then apply a "rounding procedure" $H : \mathcal{K} \to \Delta(K^\epsilon)$ that converts the recommendations to the desired form.

This algorithmic template can be described as follows. For each family of loss functions $\mathcal{L}$ we will specify:

- a discretization $K^\epsilon$: this can be an $\epsilon$-net or an $\epsilon$-triangulation of $\mathcal{K}$.

- a rounding procedure $H : \mathcal{K} \to \Delta(K^\epsilon)$ which maps each point in the original set to a distribution of points in the discretization

- a scaled external-regret-minimizing algorithm $\mathcal{A}$ for online learning problem on the original set $\mathcal{K}$ and loss functions in $\mathcal{L}$.

With those components, we can run our main algorithm for full-swap-regret-minimization. We create an instance $\text{Alg}_s$ of the external-regret algorithm $\mathcal{A}$ for each $s \in K^\epsilon$. At each round $t$, we first query each $\text{Alg}_s$ to obtain a recommended action $q_{s,t} \in \mathcal{K}$. Now, we use the rounding procedure to convert $q_{s,t} \in \mathcal{K}$ into a distribution over discretized actions $\mathbf{z}_{s,t} = H(q_{s,t}) \in \Delta(K^\epsilon)$. We then build a Markov chain with states corresponding to $K^\epsilon$ where the transition from $s$ to $s'$ is given by $\mathbf{z}_{s,t}[s']$. The learner then plays a stationary distribution $\mathbf{x}_t$ of the Markov chain. After the loss $\ell_t$ is observed, we feed each algorithm $\text{Alg}_s$ the loss $\ell_t$ and scale $g_{t,s} = \mathbf{x}_t[s]$.

Following the Blum-Mansour analysis and bounding the discretization error, we obtain the following:

**Theorem 12** *Say we have a rounding procedure $H : \mathcal{K} \to \Delta(K^\epsilon)$ such that, for all $q \in \mathcal{K}$, for all $\ell \in \mathcal{L}$, we have $\mathbb{E}_{s \sim H(q)}[\ell(s)] - \ell(q) \leq \delta$ for some $\delta > 0$. Also let $\text{ExtReg}_s$ be the scaled external regret incurred by $\text{Alg}_s$, then, for Algorithms 1 and 2: $\text{FullSwapReg} \leq \delta T + \sum_{s \in K^\epsilon} \text{ExtReg}_s$.*

See Appendix B for proof and details.

## 5. Full Swap Regret

We now instantiate the template to obtain the FullSwapReg guarantees. In Table 5, we present a summary of our results, including FullSwapReg guarantees for settings where the loss functions are strongly convex, concave, or not having any convexity/concavity assumption. Our template is designed for the regime where losses are strongly convex, and later nearly-strongly-convex. These are the settings of interest for establishing algorithms for calibration, and the settings where we stand to gain the most by utilizing the embedded structure of the actions into Euclidean space. For the concave and no-assumption settings, we are simply using the Blum Mansour framework over a discretization of the action space as a black box. We include the results for all these settings in one unified table so as to paint a complete baseline picture of what rates can be attained under this new FullSwapReg benchmark.

For external regret algorithms in the strongly convex and nearly-strongly-convex settings, we will use the variants of online gradient descent in Lemma 24 for strongly convex functions and in Theorem 20 for nearly strongly convex (we defer discussion of this latter setting to the next section).

For the discretization and rounding in the non-smooth loss case, we will use either $\epsilon$-nets in Lemma 8 coupled with the rounding $H : \mathcal{K} \to \Delta(K^\epsilon)$ that (deterministically) maps a point in $K^\epsilon$ to the nearest point in $K^\epsilon$ (i.e. the projection map). The rounding satisfies the following guarantee that trivially follows from Lipschitzness.

**Lemma 13** *For all $q \in \mathcal{K}$, for all $L$-Lipschitz loss functions $\ell : \mathcal{K} \to \mathbb{R}$, letting $H : \mathcal{K} \to \Delta(K^\epsilon)$ be the projection map on $\epsilon$-net $K^\epsilon$, we have $\mathbb{E}_{s \sim H(q)}[\ell(s)] - \ell(q) \leq L\epsilon$.*

Note that the expectation in the statement is vacuous since the projection is a deterministic map, but we write to highlight that the template allows for randomized maps.

For the discretization and rounding in the $\beta$-smooth loss case, we will use the $\epsilon$-triangulation in Lemma 10 together with the rounding procedure $H : \mathcal{K} \to \Delta(K^\epsilon)$ that takes $x \in \mathcal{K}$, projects it to $\mathrm{conv}(K^\epsilon)$ obtaining $x_\perp = \Pi_{\mathrm{conv}(K^\epsilon)}(x)$ and then outputs a probability distribution supported in the points $K^\epsilon(x_\perp)$ that equal $x_\perp$ in expectation. (Recall from Lemma 10 that in a triangulation $K^\epsilon(x)$ is a set of at most $d + 1$ points containing $x$ in the convex hull.) See Algorithm 4 in the appendix for a detailed description in pseudocode together with the proof of the following lemma. With the stronger smoothness property together with the more structured discretization, we can improve the rounding error from $O(\epsilon)$ to $O(\epsilon^2)$:

**Lemma 14** *For all $q \in \mathcal{K}$, for all L-Lipschitz, $\beta$-smooth loss functions $\ell : \mathcal{K} \to \mathbb{R}$, letting $H : \mathcal{K} \to \Delta(K^\epsilon)$ be the map induced by Algorithm 4 on $\epsilon$-triangulation $K^\epsilon$, we have $\mathbb{E}_{s \sim H(q)}\left[\ell(s)\right] - \ell(q) \leq (L + \beta/8)\,\epsilon^2$.*

For the discretization in the concave loss case, rather than taking an $\epsilon$-net of the entire set $\mathcal{K}$, it will only be necessary to take an $\epsilon$-net of the boundary. Because losses are concave, the loss-minimizing actions will always lie at extreme points in $\mathcal{K}$. It will always be preferable to represent any interior point as a convex combination of boundary points. This discretization is accomplished by taking a polytope approximation of $\mathcal{K}$. Theorem 22 (Appendix A) guarantees the existence of a polytope $\mathcal{P}$ contained entirely in $\mathcal{K}$ with at most $(1/\epsilon)^{d-1}$ vertices such that $\max_{x \in \mathcal{K}} \mathrm{dist}(x, \mathcal{P}) \leq \epsilon^2$. Taking $K^\epsilon$ to be the vertex set of $\mathcal{P}$, we shave a $(1/\epsilon)$ factor off the magnitude of $K^\epsilon$, which for the $\epsilon$-net contained $(1/\epsilon)^d$ many points as it also had to cover the interior of $\mathcal{K}$. Additionally, we maintain the improved rounding error of $O(\epsilon^2)$ from the $\beta$-smooth case as the projection of any $x \in \mathcal{K}$ to $\mathrm{conv}(K^\epsilon) = \mathcal{P}$ is guaranteed to be at most $\epsilon^2$ away from $x$.

Our main theorem is obtained from putting these parts together and optimizing the discretization parameter $\epsilon$. We state the full result below:

**Theorem 15** *Let $\mathcal{K} \subseteq \mathbb{R}^d$ be a convex set of diameter 1. Let $\mathcal{L} \subseteq \{\ell : \mathcal{K} \to \mathbb{R}\}$ be a family of L-Lipschitz loss functions. Algorithms 1 and 2 attain the full-swap-regret guarantees described in Table 5 in terms of the indicated additional constraints on the loss functions $\ell \in \mathcal{L}$, using the indicated subroutines $\mathcal{A}$, discretizations $K^\epsilon$, and rounding algorithms $H$. All algorithms run in per-iteration time $\mathrm{poly}(|K^\epsilon|) = (1/\epsilon)^{O(d)} = \mathrm{poly}(d, T)$.*

## 6. Discretized Swap Regret

We now consider a setting where a discrete set of points $K^\epsilon \subset \mathcal{K}$ is given to the learner, who is then constrained to only play randomized actions supported on this discretization $K^\epsilon$. Accordingly, the learner must only compete with swap functions $\phi : K^\epsilon \to K^\epsilon$ rather than all transformations $\phi : K^\epsilon \to \mathcal{K}$. While this is a special case of classic online learning over finitely many actions $k = |K^\epsilon|$, under certain assumptions about the discretization $K^\epsilon$ and the loss functions $\ell_t$, this additional structure enables an improved swap regret bound. As a consequence, we obtain improved bounds for discretized calibration (see Figure 1).

We will again treat this as a problem over the convex hull $\mathrm{conv}(K^\epsilon)$ and we will round points to a distribution of points in $K^\epsilon$. The main tool we will use is to replace the loss function $\ell_t$ with its piecewise-linearization $\dot{\ell}_t$ which we describe below. All the key definitions make sense for any

Table 1: Matrix of regret bounds for Theorem 15.

| Extra assumptions on $L$-Lipschitz losses $\ell$ | FullSwapReg Algorithm and ExtReg Subroutine | Discretization and rounding | FullSwapReg Rate |
|---|---|---|---|
| $\ell$: no assumption | Algorithm 2 $\mathcal{A}$: MWU | $K^\epsilon$: $\epsilon$-net, $\epsilon = T^{-1/(d+2)}$ | $O\left(LT^{\frac{d+1}{d+2}} \log T\right)$ |
| $\ell$: $\beta$-smooth | Algorithm 2 $\mathcal{A}$: MWU | $K^\epsilon$: $\epsilon$-triangulation, $\epsilon = T^{-1/(d+4)}$ | $O\left((L+\beta)T^{\frac{d+2}{d+4}} \log T)\right)$ |
| $\ell$: linear or concave | Algorithm 2 $\mathcal{A}$: MWU | $K^\epsilon$: polytope approximation, $\epsilon = T^{-1/(d+3)}$ | $O\left(LT^{\frac{d+1}{d+3}} \log T\right)$ |
| $\ell$: $\alpha$-strongly-convex | Algorithm 1 $\mathcal{A}$: Algorithm 3 | $K^\epsilon$: $\epsilon$-net, $\epsilon = (L/\alpha)^{\frac{1}{d+1}}T^{-\frac{1}{d+1}}$ $H$: Projection | $O\left(L\left(\frac{L}{\alpha}\right)^{\frac{1}{d+1}} T^{\frac{d}{d+1}} \log T\right)$ |
| $\ell$: $\alpha$-strongly-convex and $\beta$-smooth | Algorithm 1 $\mathcal{A}$: Algorithm 3 | $K^\epsilon$: $\epsilon$-triangulation, $\epsilon = T^{-1/(d+2)}$ $H$: Algorithm 4 | $O\left(L\left(\frac{\beta}{L} + \frac{L}{\alpha}\right) T^{\frac{d}{d+2}} \log T\right)$ |

dimension $d$ but we are only able to bound the performance of the rounding procedure for $d = 1$. We leave as an open question how to extend the analysis of the rounding procedure to higher dimensions. We keep the presentation general in the hope that some of those tools will be of independent interest.

**Definition 16 (Piecewise-linearization)** *For a set $K^\epsilon \subseteq \mathbb{R}^d$ with $|K^\epsilon| = k$, and a convex loss function $\ell : \text{conv}(K^\epsilon) \to \mathbb{R}$, we define the piecewise-linearized loss function $\dot{\ell}_{K^\epsilon}$ to be the lower envelope of the convex hull $\text{conv}\{(s, \ell(s))|s \in K^\epsilon\}$. Equivalently, for all $\mathbf{x} \in \text{conv}(K^\epsilon)$: $\dot{\ell}_{K^\epsilon}(\mathbf{x}) = \min_{v \in \Delta(K^\epsilon); \mathbb{E}[v]=\mathbf{x}} \langle v, \mu(K^\epsilon) \rangle$ where $\mu(K^\epsilon) \in \mathbb{R}^k$ denotes the vector with entries $\ell(s)$ for each $s \in K^\epsilon$. We abbreviate $\dot{\ell} = \dot{\ell}_{K^\epsilon}$ when $K^\epsilon$ is clear from context.*

*For the special case where $K^\epsilon = \{s_1, \cdots, s_k\} \subset \mathbb{R}$ with $s_i < s_{i+1}$ for all $i$, we can simplify this expression. For all $x \in \text{conv}(K^\epsilon) = [s_1, s_k]$, letting $i(x) \in [k]$ satisfy $\mathbf{x} \in [s_i, s_{i+1}]$, $\dot{\ell}(x) = \frac{\ell(s_{i(x)+1}) - \ell(s_{i(x)})}{s_{i(x)+1} - s_{i(x)}}(x - s_{i(x)}) + \ell(s_{i(x)})$.*

For the case where $d = 1$, we utilize the simple rounding procedure:

$$H(x)[s_{i(x)}] = \frac{s_{i(x)+1} - x}{s_{i(x)+1} - s_{i(x)}}, \quad H(x)[s_{i(x)+1}] = \frac{x - s_{i(x)}}{s_{i(x)+1} - s_{i(x)}}, \quad H(x)[s_i] = 0 \text{ o.w.} \quad (1)$$

We will use the following fact about this rounding procedure which is special to $d = 1$.

**Lemma 17** *Let $K^\epsilon \subset \mathbb{R}$. For all $t \in [T]$, for recommendations $x_t \in \text{conv}(K^\epsilon)$, and convex loss functions $\ell_t : \text{conv}(K^\epsilon) \to \mathbb{R}$. We have $\dot{\ell}_t(x_t) = \mathbb{E}_{s \sim H(x_t)}[\ell_t(s)]$.*

**Proof** This holds since the mixture $H(x)$ is supported on the interval $[s_{i(x)}, s_{i(x)+1}]$ over which $\dot{\ell}$ is linear. Thus, $\dot{\ell}_t(x_t) = \mathbb{E}_{s \sim H(x_t)}[\dot{\ell}_t(s)] = \mathbb{E}_{s \sim H(x_t)}[\ell_t(s)]$. ∎

For higher dimensions, it is not possible to find a 'lossless' rounding before seeing the actual loss function.

Following our algorithmic template, we need external-regret-minimization algorithms for loss functions resulting from piecewise-linearization of $\alpha$-strongly convex functions. Unfortunately they are no longer $\alpha$-strongly convex, but they are still what we call $(\alpha, \epsilon)$-nearly-strongly convex.

**Definition 18** *Consider a convex set $\mathcal{K} \subseteq \mathbb{R}^d$. We say that a continuous function $\ell : \mathcal{K} \to \mathbb{R}$ is $(\alpha, \epsilon)$-**nearly strongly convex** if, for all $\mathbf{x}, \mathbf{y} \in \mathcal{K}$*

$$\ell(\mathbf{y}) - \ell(\mathbf{x}) - \nabla\ell(\mathbf{x})^\top (\mathbf{y} - \mathbf{x}) \geq \frac{\alpha}{2} \left( \|\mathbf{y} - \mathbf{x}\| - \epsilon \right)_+^2 \tag{2}$$

*where $(x)_+ = \max(x, 0)$ and $\nabla\ell(\mathbf{x})$ can be any subgradient of $\mathbf{x}$.*

We establish the following lemma demonstrating that piecewise-linearizing strongly-convex loss functions results in nearly-strongly-convex loss functions (proof deferred to [Appendix D.6](#)).

**Lemma 19** *Let $K^\epsilon = \{s_1, \cdots, s_k\} \subset \mathbb{R}$ be a set of reals with $s_i < s_{i+1}$ for all $i$. Let $K^\epsilon$ be an $\epsilon$-triangulation of $\mathrm{conv}(K^\epsilon)$ (i.e. $s_{i+1} - s_i \leq \epsilon$ for all $i$). Let $\ell : \mathrm{conv}(K^\epsilon) \to \mathbb{R}$ be $\alpha$-strongly-convex. Then, $\dot{\ell}$ is $(\alpha, \epsilon)$-nearly-strongly-convex, where $\dot{\ell}$ is the piecewise-linearized $\ell$ according to Definition [16](#).*

**External Regret Minimization for Nearly Strongly Convex Functions** We have reduced the task of external regret minimization of strongly convex loss functions over a discretization $K^\epsilon$ to external regret minimization of nearly-strongly-convex loss functions over the set $\mathrm{conv}(K^\epsilon)$. However, existing algorithms don't work out of the box. One option would be to pretend the nearly-strongly-convex loss functions are actually strongly convex, and apply Algorithm [3](#). The total regret of such algorithm would be $O(\log T + \epsilon T)$ where $\epsilon T$ is due to the fact that these nearly-strongly-convex loss functions may be linear over intervals of length $\epsilon$. Similarly we could treat them as standard convex functions, not taking advantage of the of the strong convexity for distant points, and obtain $O(\sqrt{T})$ bound which is again not sufficient.

A key technical contribution of this work is to show that we can apply OGD with a learning rate schedule that initially decays with $1/t$ and later switches to decay $1/\sqrt{t}$ at the right moment. This allows us to capture the effect of strong-convexity for points that are far enough from the optimum, for which the function behaves like a strongly convex function. At the same time, it allows OGD to treat the function as a standard Lipschitz function for points close to the optimum.

Formally, the scaled version of OCO with losses $\ell_t$ and scales $g_t$, the algorithm will apply the standard OGD update: $\mathbf{x}_{t+1} \leftarrow \Pi_\mathcal{K} \left( \mathbf{x}_t - \eta_t g_t \nabla \ell_t(\mathbf{x}_t) \right)$ with the learning rate $\eta_t = R'(G_t)$ where $G_t = \sum_{s=1}^t g_s$ for the function:

$$R'(x) = \frac{2}{\alpha} \text{ for } x \in [0, 1], \quad R'(x) = \frac{2}{\alpha x} \text{ for } x \in \left[ 1, \left( \frac{\sqrt{2}L}{\alpha \epsilon} \right)^2 \right], \quad R'(x) = \frac{\sqrt{2}\epsilon}{L\sqrt{x}} \text{ for } x \geq \left( \frac{\sqrt{2}L}{\alpha \epsilon} \right)^2$$

**Theorem 20** *For $(\alpha, \epsilon)$-nearly-strongly-convex loss functions $\ell_t$ and scale parameters $g_t$, an instance of Online Gradient Descent (Algorithm 5 in the appendix) achieves the following scaled external regret guarantee:*

$$\mathsf{ExtReg} \leq 2\sqrt{2}\epsilon L \sqrt{G_T} + \frac{L^2}{\alpha} \left( \log(G_T + 1) + 1 \right)$$

### 6.1. Swap Regret and Discretized $\ell_2$-calibration

We now can use the algorithm in Theorem 20 together with the lossless rounding procedure in Lemma 17 in our algorithmic template to obtain the following bound:

**Theorem 21** *Let $K^\epsilon \subset \mathbb{R}$ be a set of reals such that $K^\epsilon$ is an $\epsilon$-net of $\mathrm{conv}(K^\epsilon)$ and $\mathrm{conv}(K^\epsilon)$ has diameter 1. Let $\ell_t$ be $\alpha$-strongly-convex, $L$-Lipschitz loss functions. Let $H$ be the rounding procedure described in (1). Let $\mathcal{A}$ be the external-regret-minimizing subroutine Algorithm 6.*
*Then the $\mathsf{FullSwapReg}$ achieved with respect to the discretized set $K^\epsilon$ satisfies:*

$$\mathsf{FullSwapReg} = O\left( L\sqrt{\epsilon T} + \frac{L^2}{\alpha\epsilon} \log(T) \right)$$

*For the regime $\frac{1}{\epsilon} = o(\sqrt{T})$, we improve on Theorem 15 for non-smooth losses, and for the regime $\frac{1}{\epsilon} = o(T^{1/3})$, we improve on Theorem 15 for $\beta$-smooth losses.*

Theorem 21 can be directly applied to our formulation of $\ell_2$-calibration as swap regret to prove our $O(\max(\sqrt{\epsilon T}, T^{1/3}))$ regret bound for $\epsilon$-discretized calibration (Theorem 5). This is deferred to Appendix D.4.

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

## Appendix A. Existence of triangulations

In this appendix, we provide a self-contained proof that sufficiently good triangulations exist, establishing Lemma 10. Our main tool will be the following theorem of Bronshteyn and Ivanov (1975) on approximations of convex sets by polytopes.

**Theorem 22 (Bronshteyn and Ivanov (1975))** *Let $\mathcal{K}$ be any $d$-dimensional convex set contained in the unit ball. Then, for any $\delta < 0.01$, there exists a polytope $\mathcal{P}$ contained in $\mathcal{K}$ with at most $O(\sqrt{d} \cdot \delta^{-(d-1)/2})$ vertices such that $\max_{x \in \mathcal{K}} \text{dist}(x, \mathcal{P}) \leq \delta$.*

To form a good triangulation of $\mathcal{K}$, we will apply Theorem 22 to the epigraph of a quadratic function over $\mathcal{K}$. The epigraph of a convex function $f : \mathcal{K} \to \mathbb{R}$ is defined to be the $(d + 1)$-dimensional convex set $\text{graph}(f) = \{(x, y) \mid x \in \mathcal{K}, y \geq f(x)\}$. The following lemma relates the closest distance between a point and the epigraph of $f$ to the vertical distance between the point and the epigraph of $f$.

**Lemma 23** *Let $\mathcal{K}$ be a $d$-dimensional convex set contained in the unit ball and let $f : \mathcal{K} \to \mathbb{R}$ be a convex function with bounded subgradient $\|\nabla f(x)\| \leq G$. Then given any point $p = (x, y) \in \mathbb{R}^{d+1}$ with $y \leq f(x)$, we have that*

$$f(x) - y \leq \sqrt{1 + G^2}\, \text{dist}(p, \text{graph}(f)).$$

**Proof** Consider any hyperplane $H$ tangent to $f$ at $x$. This hyperplane separates the point $p$ from the graph of the function $f$, so $\text{dist}(p, H) \leq \text{dist}(p, \text{graph}(f))$. On the other hand, the distance from $p$ to $H$ is given by

$$\text{dist}(p, H) = \frac{f(x) - y}{\sqrt{1 + \|\nabla f(x)\|^2}}.$$

Since $\|\nabla f(x)\| \leq G$, the result follows. ∎

We can now prove our main lemma.

**Proof** [Proof of Lemma 10]

We will handle both constraints of the definition of an $\epsilon$-triangulation by separate applications of Theorem 22. We start with the first constraint (that all elements on the boundary of $\mathcal{K}$ must be within $\epsilon^2$ of $\text{conv}(K^\epsilon)$). To do this, note that we can directly apply Theorem 22 to the convex set $\mathcal{K}$ (with parameter $\delta = \epsilon^2$) to obtain a set $K_1$ of at most $O(\sqrt{d}\epsilon^{-(d-1)})$ points with the property that $\max_{x \in \mathcal{K}} \text{dist}(x, \text{conv}(K_1)) \leq \epsilon^2$.

This construction is only guaranteed to well approximate the boundary of $\mathcal{K}$. To approximate the interior of $\mathcal{K}$ and satisfy the second condition, let $f : \text{conv}(K_1) \to \mathbb{R}$ be the convex function $f(x) = \|x\|^2$. Consider the convex set $\mathcal{K}' = \text{graph}(f) \cap \{y \leq 1\}$ formed by truncating the epigraph of $f$ for all $y \geq 1$ (note that since $\|x\| \leq 1$ for $x \in \mathcal{K}$, this does not remove any of the vertices of the form $(x, f(x))$). Since $f(x) \leq 1$ for $(x, y) \in \mathcal{K}$, the set $\mathcal{K}'$ is contained within a ball of radius $\sqrt{2}$, so by applying Theorem 22 to a scaled version of $\mathcal{K}'$ (with parameter $\delta = \epsilon^2$) we obtain a set $K_2$ containing at most $O(\sqrt{d}\epsilon^{-d})$ points in $\mathcal{K}'$ with the property that $\max_{x \in \mathcal{K}'} \text{dist}(x, \text{conv}(K_2)) \leq 2\epsilon^2$. Now, without loss of generality we will make two modifications to $K_2$:

- First, for each vertex $x \in K_1$, we will assume that $(x, 1) \in K_2$. If not, we can just add all these points to $K_2$ (this doesn't affect the asymptotic number of points in $K_2$).

- For all other points $(x, y) \in K_2$, we assume that $y = f(x)$ (i.e., this point lies on the boundary of $K_2$). If not, note that replacing it with $(x, f(x))$ strictly increases the size of $\mathrm{conv}(K_2)$ (in particular, $(x, y)$ is a convex combination of $(x, f(x))$ and $(x, 1)$, and the latter point belongs to $\mathrm{conv}(K_2)$ by the previous bullet).

We choose $K^\epsilon$ to be the projection of $K_2$ onto the first $d$ coordinates. To see that this satisfies the second constraint, consider the lower envelope of $\mathrm{conv}(K_2)$; that is, the function $g : \mathrm{conv}(K_1) \to \mathbb{R}$ defined so that $g(x)$ equals the minimum $y$ such that $(x, y) \in \mathrm{conv}(K_2)$ (for this $g$, $\mathrm{graph}(g) \cap \{y \leq 1\} = \mathrm{conv}(K_2)$). Note that the subgradient of $g$ is bounded by the maximum subgradient of $f$ over $\mathrm{conv}(K_1)$. The gradient $\nabla f(x) = 2 \cdot ||x|| \leq 2$, so $||\nabla g(x)|| \leq 2$.

Now, consider any $x \in \mathrm{conv}(K_1)$ and let $p = (x, f(x))$. By the guarantees of Theorem 22, $\mathrm{dist}(p, \mathrm{graph}(g)) = \mathrm{dist}(p, \mathrm{conv}(K_2)) \leq \epsilon^2$. By Lemma 23, this implies that $g(x) - f(x) \leq \sqrt{5}\epsilon^2$. But $(x, g(x))$ also lies on some $d$-dimensional face of $K_2$ and can be written as a convex hull of at most $d + 1$ points in $K_2$ of the form $(x_i, f(x_i))$. We will choose $K^\epsilon(x)$ to be this set of $x_i$. Write $x = \sum_i \lambda_i x_i$; then $g(x) = \sum_i \lambda_i f(x_i) = \sum_i \lambda_i ||x_i||^2$. We can now express

$$
\begin{aligned}
g(x) - f(x) &= \sum_i \lambda_i ||x_i||^2 - ||x||^2 \\
&= \sum_i \lambda_i ||x + (x_i - x)||^2 - ||x||^2 \\
&= \sum_i \lambda_i ||x_i - x||^2
\end{aligned}
$$

In particular, substituting $x = \frac{1}{2}x_i + \frac{1}{2}x_j$ for $i \neq j$ above and using $g(x) - f(x) \leq \sqrt{5}\epsilon^2$, we obtain: $\frac{1}{2} ||x_i - x_j||^2 \leq \sqrt{5}\epsilon^2$. It follows that for any $x_i, x_j \in K^\epsilon(x)$, $||x_i - x_j|| \leq O(\epsilon)$. It follows that $\mathrm{diam}(K^\epsilon(x)) \leq O(\epsilon)$, as desired. ∎

## Appendix B. Analysis of the Algorithmic Template

Here, we detail our main algorithmic templates for swap regret minimization and prove Theorem 12. We begin with our algorithm for strongly convex and nearly-strongly-convex losses. We first re-state the algorithm in pseudocode:

Note that we pass as input to the black-box external-regret-minimizing instance $\mathtt{Alg}_s$ both the loss $\ell_t$ as well as the weight $g_t = \mathbf{x}_t[s]$ placed on the corresponding action that round. The loss incurred by this instance will be $g_t \ell_t = \mathbf{x}_t[s]\ell_t$: the loss scaled by this weight. We represent this as two separate inputs as it will be notationally convenient later when proving that the instance achieves a "first-order" external-regret bound in terms of $\sum g_t$.

Now, we provide pseudocode for our algorithmic template in the event that our losses are non-convex. Here, we just rely on a black-box implementation of Blum Mansour over a discretization of $\mathcal{K}$:

---

**Algorithm 1** Blum Mansour for Convex Sets and Convex losses

---

**Input:** Convex set $\mathcal{K} \subseteq \mathbb{R}^d$, diameter 1
**Input:** Loss family $\mathcal{L} \subseteq \{\ell : \mathcal{K} \to \mathbb{R}\}$
**Input:** $K^\epsilon$: discretization of $\mathcal{K}$
**Input:** $H$: rounding procedure $\mathcal{K} \to \Delta(K^\epsilon)$
**Input:** External-regret-minimizing algorithm $\mathcal{A}$; outputs to $\mathcal{K}$
**Input:** For all $t \in [T]$: $\ell_t \in \mathcal{L}$ revealed sequentially
**Output:** For all $t \in [T]$: strategies $\mathbf{x}_t \in \Delta(K^\epsilon)$
**for** $s \in K^\epsilon$ **do**
$\quad$| Instantiate copy of $\mathcal{A}$: $\texttt{Alg}_s$
**end**
**for** $t = 1 : T$ **do**
$\quad$**for** $s \in K^\epsilon$ **do**
$\quad\quad$| $q_{s,t} \leftarrow \texttt{Alg}_s$ recommendation at time $t$
$\quad\quad$| $\mathcal{Q}_t[s] \leftarrow H(q_{s,t})$
$\quad\quad$| // Setting row $s$ of a $K^\epsilon \times K^\epsilon$ row-stochastic matrix
$\quad$**end**
$\quad$$\mathbf{x}_t \leftarrow$ stationary distribution$(\mathcal{Q}_t)$
$\quad$**return** $\mathbf{x}_t$
$\quad$**observe** $\ell_t$
$\quad$**for** $s \in K^\epsilon$ **do**
$\quad\quad$| **update** $\texttt{Alg}_s$ with $\ell_t$ and $g_t \leftarrow \mathbf{x}_t[s]$
$\quad$**end**
**end**

---

---

**Algorithm 2** Blum Mansour for Convex Sets and Non-Convex losses

---

**Input:** Convex set $\mathcal{K} \subseteq \mathbb{R}^d$, diameter 1
**Input:** Loss family $\mathcal{L} \subseteq \{\ell : \mathcal{K} \to \mathbb{R}\}$
**Input:** $K^\epsilon$: discretization of $\mathcal{K}$
**Input:** External-regret-minimizing algorithm $\mathcal{A}$; outputs to $\Delta(K^\epsilon)$
**Input:** For all $t \in [T]$: $\ell_t \in \mathcal{L}$ revealed sequentially
**Output:** For all $t \in [T]$: strategies $\mathbf{x}_t \in \Delta(K^\epsilon)$
**for** $s \in K^\epsilon$ **do**
  | Instantiate copy of $\mathcal{A}$: $\texttt{Alg}_s$
**end**
**for** $t = 1$ **to** $T$ **do**
  | **for** $s \in K^\epsilon$ **do**
  |   | $\mathbf{q}_{s,t} \leftarrow \texttt{Alg}_s$ recommendation at time $t$
  |   | $\mathcal{Q}_t[s] \leftarrow \mathbf{q}_{s,t}$
  |   | `// Setting row s of a `$K^\epsilon \times K^\epsilon$` row-stochastic matrix`
  | **end**
  | $\mathbf{x}_t \leftarrow$ stationary distribution$(\mathcal{Q}_t)$
  | **return** $\mathbf{x}_t$
  | **observe** $\ell_t$
  | **for** $s \in K^\epsilon$ **do**
  |   | **update** $\texttt{Alg}_s$ with $\ell_t$ and $g_t \leftarrow \mathbf{x}_t[s]$
  | **end**
**end**

---

We are ready to prove our main theorem about the full-swap-regret achieved by Algorithms 1 and 2, restated below. We will later use this theorem as a black box to demonstrate that the algorithm also achieves good discretized swap regret for particular settings of $\mathcal{L}$.

**Theorem 12** Say we have a rounding procedure $H : \mathcal{K} \to \Delta(K^\epsilon)$ such that, for all $q \in \mathcal{K}$, for all $\ell \in \mathcal{L}$, we have $\mathbb{E}_{s \sim H(q)}[\ell(s)] - \ell(q) \leq \delta$ for some $\delta > 0$. Also let $\mathsf{ExtReg}_s$ be the scaled external regret incurred by $\mathtt{Alg}_s$, then, for Algorithms 1 and 2: $\mathsf{FullSwapReg} \leq \delta T + \sum_{s \in K^\epsilon} \mathsf{ExtReg}_s$.

**Proof** [Proof of Theorem 12] First, for Algorithm 1, we have

$\mathsf{FullSwapReg}\,(\mathbf{x}_{1:T}, \ell_{1:T})$

$$= \sum_{s \in \mathcal{K}} \max_{\phi(s) \in \mathcal{K}} \left( \sum_{t=1}^{T} \mathbf{x}[s](\ell_t(s) - \ell_t(\phi(s))) \right)$$

$$= \sum_{s \in K^\epsilon} \max_{\phi(s) \in \mathcal{K}} \left( \sum_{t=1}^{T} \mathbf{x}[s](\ell_t(s) - \ell_t(\phi(s))) \right)$$

$$= \sum_{s \in K^\epsilon} \sum_{t=1}^{T} \left( \sum_{s' \in K^\epsilon} \mathbf{x}[s']\mathcal{Q}[s'][s] \right) \ell_t(s) - \left( \sum_{s \in K^\epsilon} \max_{\phi(s) \in \mathcal{K}} \sum_{t=1}^{T} \mathbf{x}[s]\ell_t(\phi(s)) \right)$$

$$= \sum_{s' \in K^\epsilon} \sum_{t=1}^{T} \mathbf{x}[s']\mathbb{E}_{s \sim H(q_{s',t})}[\ell_t(s)] - \left( \sum_{s \in K^\epsilon} \max_{\phi(s) \in \mathcal{K}} \sum_{t=1}^{T} \mathbf{x}[s]\ell_t(\phi(s)) \right)$$

$$\leq \sum_{s' \in K^\epsilon} \sum_{t=1}^{T} \mathbf{x}[s'] \left( \ell_t(q_{s',t}) + \delta \right) - \left( \sum_{s \in K^\epsilon} \max_{\phi(s) \in \mathcal{K}} \sum_{t=1}^{T} \mathbf{x}[s]\ell_t(\phi(s)) \right)$$

$$= \delta T + \sum_{s \in K^\epsilon} \max_{\phi(s) \in \mathcal{K}} \sum_{t=1}^{T} (\mathbf{x}[s]\ell_t(q_{s,t}) - \mathbf{x}[s]\ell_t(\phi(s)))$$

$$= \delta T + \sum_{s \in K^\epsilon} \mathsf{ExtReg}\,(q_{s,1:T}, \mathbf{x}[s]\ell_{1:T})$$

as desired. For Algorithm 2, we have

$$\mathsf{FullSwapReg}\,(\mathbf{x}_{1:T}, \ell_{1:T})$$

$$= \sum_{s \in \mathcal{K}} \max_{\phi(s) \in \mathcal{K}} \left( \sum_{t=1}^{T} \mathbf{x}[s](\ell_t(s) - \ell_t(\phi(s))) \right)$$

$$= \sum_{s \in K^{\epsilon}} \max_{\phi(s) \in \mathcal{K}} \left( \sum_{t=1}^{T} \mathbf{x}[s](\ell_t(s) - \ell_t(\phi(s))) \right)$$

$$= \sum_{s \in K^{\epsilon}} \sum_{t=1}^{T} \left( \sum_{s' \in K^{\epsilon}} \mathbf{x}[s']\mathcal{Q}[s'][s] \right) \ell_t(s) - \left( \sum_{s \in K^{\epsilon}} \max_{\phi(s) \in \mathcal{K}} \sum_{t=1}^{T} \mathbf{x}[s]\ell_t(\phi(s)) \right)$$

$$= \sum_{s' \in K^{\epsilon}} \sum_{t=1}^{T} \mathbf{x}[s']\ell_t(q_{s',t}) - \left( \sum_{s \in K^{\epsilon}} \max_{\phi(s) \in \mathcal{K}} \sum_{t=1}^{T} \mathbf{x}[s]\ell_t(\phi(s)) \right)$$

$$= \sum_{s \in K^{\epsilon}} \max_{\phi(s) \in \mathcal{K}} \sum_{t=1}^{T} (\mathbf{x}[s]\ell_t(q_{s,t}) - \mathbf{x}[s]\ell_t(\phi(s)))$$

$$= \sum_{s \in K^{\epsilon}} \max_{\phi(s) \in \mathcal{K}} \sum_{t=1}^{T} (\mathbf{x}[s]\ell_t(q_{s,t}) - \mathbf{x}[s]\mathbb{E}_{s^* \sim H(\phi(s))}\left[\ell_t(s^*)\right]$$

$$+ \mathbf{x}[s]\mathbb{E}_{s^* \sim H(\phi(s))}\left[\ell_t(s^*)\right] - \mathbf{x}[s]\ell_t(\phi(s)))$$

$$\leq \delta T + \sum_{s \in K^{\epsilon}} \mathsf{ExtReg}\,(q_{s,1:T}, \mathbf{x}[s]\ell_{1:T})$$

as desired. ∎

## Appendix C. External Regret Minimization Algorithms

We present the external-regret-minimizing algorithm $\mathcal{A}$ for use as a subroutine in Algorithm 1 when the loss functions are $\alpha$-strongly-convex. As stated previously, this algorithm receives two inputs at each time step $t$: loss function $\ell_t$ and scale parameter $g_t$. When evaluating the external regret of these algorithms, we do so in terms of the scaled loss functions $g_t \ell_t$. They are input separately merely for notational convenience. We will derive regret bounds in terms of the first-order quantity $G_T = \sum_{t=1}^{T} g_t$.

When we assume the loss functions in $\mathcal{L}$ are $\alpha$-strongly-convex and $L$-Lipschitz, we use the following external-regret-minimizing algorithm. Define

$$R'_{\alpha}(x) := \begin{cases} \frac{1}{\alpha} & \text{for } x \in [0, 1] \\ \frac{1}{\alpha x} & \text{for } x \geq 1 \end{cases}$$

We abbreviate $R' := R'_{\alpha}$ for convenience.

**Lemma 24** *Let $\mathcal{K} \subseteq \mathbb{R}^d$ be a convex set of diameter $\leq 1$. Let*

$$\mathcal{L} = \{\ell : \mathcal{K} \to \mathbb{R} | \ell: \alpha\text{-strongly-convex and } L\text{-Lipschitz}\}$$

---

**Algorithm 3** Online Gradient Descent for Strongly Convex Loss

---

**Input:** Convex set $\mathcal{K} \subseteq \mathbb{R}^d$
**Input:** For all $t \in [T]$: $\alpha$-strongly-convex, $L$-Lipschitz loss functions $\ell_t : \mathcal{K} \to \mathbb{R}$
**Input:** For all $t \in [T]$: Scale parameters $g_t \in [0, 1]$
**Input:** Initial $\mathbf{x}_1 \in \mathcal{K}$
**Output:** For all $t \in [T]$: strategies $\mathbf{x}_t \in \mathcal{K}$
$G_0 \leftarrow 0$
**for** $t = 1$ **to** $T$ **do**
  **return** $\mathbf{x}_t$
  **observe** $\ell_t, g_t$
  $G_t \leftarrow G_{t-1} + g_t$
  $\eta_t \leftarrow R'(G_t)$
  $\mathbf{x}_{t+1} \leftarrow \Pi_{\mathcal{K}} \left( \mathbf{x}_t - \eta_t g_t \nabla \ell_t(\mathbf{x}_t) \right)$
**end**

---

*For all $T$, for all $\ell_{1:T} \in \mathcal{L}^T$, for all $g_{1:T} \in [0,1]^T$, the strategies $\mathbf{x}_{1:T}$ recommended by Algorithm 3 satisfy*

$$\mathsf{ExtReg}\left(\mathbf{x}_{1:T}, g\ell_{1:T}\right) \leq \frac{L^2}{2\alpha} \left(\log\left(G_T + 1\right) + 1\right)$$

**Proof**

Let $\nabla_t = \nabla \ell_t(\mathbf{x}_t)$. For all $\mathbf{x}^* \in \mathcal{K}$, we have $\ell_t(\mathbf{x}_t) - \ell_t(\mathbf{x}^*) \leq \langle \nabla_t, \mathbf{x}_t - \mathbf{x}^* \rangle - \frac{\alpha}{2} \|\mathbf{x}_t - \mathbf{x}^*\|^2$ by $\alpha$-strong-convexity. Also,

$$\|\mathbf{x}_{t+1} - \mathbf{x}^*\|^2 = \|\Pi_{\mathcal{K}}(\mathbf{x}_t - \eta_t g_t \nabla_t) - \mathbf{x}^*\|^2 \leq \|\mathbf{x}_t - \eta_t g_t \nabla_t - \mathbf{x}^*\|^2$$
$$= \|\mathbf{x}_t - \mathbf{x}^*\|^2 + \eta_t^2 g_t^2 \|\nabla_t\|^2 - 2\eta_t g_t \langle \nabla_t, \mathbf{x}_t - \mathbf{x}^* \rangle$$

and therefore

$$g_t \langle \nabla_t, \mathbf{x}_t - \mathbf{x}^* \rangle \leq \frac{1}{2\eta_t} \left( \|\mathbf{x}_t - \mathbf{x}^*\|^2 - \|\mathbf{x}_{t+1} - \mathbf{x}^*\|^2 \right) + \frac{\eta_t g_t^2}{2} \|\nabla_t\|^2$$
$$\leq \frac{1}{2\eta_t} \left( \|\mathbf{x}_t - \mathbf{x}^*\|^2 - \|\mathbf{x}_{t+1} - \mathbf{x}^*\|^2 \right) + \frac{\eta_t g_t^2 L^2}{2}$$

Summing over $t$

$$2 \sum_{t=1}^{T} g_t(\ell_t(\mathbf{x}_t) - \ell_t(\mathbf{x}^*)) \leq \sum_{t=1}^{T} \|\mathbf{x}_t - \mathbf{x}^*\|^2 \left( \frac{1}{\eta_t} - \frac{1}{\eta_{t-1}} - \alpha g_t \right) + L^2 \sum_{t=1}^{T} \eta_t g_t^2$$

Since $\frac{1}{\eta_t} = \alpha \max(G_t, 1)$,

$$\frac{1}{\eta_t} - \frac{1}{\eta_{t-1}} - \alpha g_t \leq \alpha \left( G_t - G_{t-1} - g_t \right) = 0$$

Additionally,

$$\sum_{t=1}^{T} \eta_t g_t^2 \leq \sum_{t=1}^{T} R'(G_t) g_t \tag{3}$$

$$= \sum_{t=1}^{T} R'(G_t)(G_t - G_{t-1})$$

$$\leq \sum_{t=1}^{T} (R(G_t) - R(G_{t-1})) = R(G_T) \tag{4}$$

where (3) follows from $g_t \leq 1$ and (4) follows from the fact that $R(x) = \int_0^x R'(x) dx$ is concave since $\frac{d}{dx} R'(x) \leq 0$. Thus,

$$\mathsf{ExtReg}\,(\mathbf{x}_{1:T}, \ell_{1:T}) = \sum_{t=1}^{T} (g_t \ell_t(\mathbf{x}_t) - g_t \ell_t(\mathbf{x}^*)) \leq \frac{L^2}{2} R(G_T)$$

$$\leq \frac{L^2}{2\alpha} \left( \log\left(G_T + 1\right) + 1 \right)$$

as desired. ∎

## Appendix D. Omitted proofs and results

### D.1. General normal-form games as structured games

Here we prove that any normal-form game with $n$ actions for the first player and $n'$ actions for the second player can be expressed as a $d$-dimensional structured game for $d = \min(n, n')$.

**Lemma 25** *Let $G$ be a normal-form game with $n$ actions for the Learner and $n'$ actions for the Adversary. Then $G$ is a $d$-dimensional structured game for $d = \min(n, n')$.*

**Proof** By symmetry, it suffices to show that the Learner's payoff matrix $u_L(i, j)$ (denoting the Learner's payoff when they play pure action $i$ and the Adversary plays pure action $j$) can be written in the form $\langle v_i, w_j \rangle$ for some $d$-dimensional vectors $v_i$ and $w_j$. If $\min(n, n') = n$, we can accomplish this by letting $v_i = e_i$ (the $i$th basis vector) and $w_j = \sum_{k=1}^{n} u_L(k, j) e_k$. Similarly, if $\min(n, n') = n'$, we can accomplish this by letting $w_j = e_j$ and $v_i = \sum_{k=1}^{n'} u_L(i, k) e_k$. ∎

### D.2. Equivalence of $\ell_2$ calibration and swap regret

Here we establish that $\ell_2$ calibration can be viewed as the swap regret in a two-dimensional structured game with infinitely many actions, as well as the full swap regret in a one-dimensional domain with strictly convex losses.

**Lemma 26 ($\ell_2$-Calibration as Swap Regret)** *For any sequences $\mathbf{x}_1, \mathbf{x}_2, \ldots, \mathbf{x}_T \in \Delta([0, 1])$ and $b_1, b_2, \ldots, b_T \in \{0, 1\}$, the following three quantities agree:*

1. *The $\ell_2$ calibration error $\mathsf{Cal}(\mathbf{x}_{1:T}, \mathbf{b}_{1:T})$.*

2. *The full swap regret $\mathsf{FullSwapReg}(\mathbf{x}_{1:T}, \ell_{1:T})$ incurred by playing the actions $\mathbf{x}_t \in \Delta([0,1])$ against the losses $\ell_t : [0,1] \to \mathbb{R}$ given by $\ell_t(x) = (x - b_t)^2$.*

3. *The swap regret $\mathsf{SwapReg}(\mathbf{x}_{1:T}, \mathbf{b}_{1:T})$ of the Learner in the repeated two-dimensional structured game where the Learner has pure actions $x \in [0,1]$ with embeddings $v_x = (2x - 1, -x^2)$ and the Adversary has pure actions $b \in \{0,1\}$ with embeddings $w_b = (b, 1)$.*

**Proof** We first show the equivalence of 1 and 2. Note that $\mathsf{FullSwapReg}$ with losses $\ell_t$ can be written as

$$
\begin{aligned}
\mathsf{FullSwapReg}(\mathbf{x}_{1:T}, \ell_{1:T}) &= \sup_{\phi:\mathcal{K}\to\mathcal{K}} \sum_{t=1}^{T} \left( \mathbb{E}_{p\sim\mathbf{x}_t}[\ell_t(p)] - \mathbb{E}_{p\sim\mathbf{x}_t}[\ell_t(\phi(p))] \right) \\
&= \sup_{\phi:\mathcal{K}\to\mathcal{K}} \sum_{t=1}^{T} \sum_{p\in\mathcal{K}} \mathbf{x}_t[p] \left( \ell_t(p) - \ell_t(\phi(p)) \right) \\
&= \sum_{p\in\mathcal{K}} \sup_{p^*\in\mathcal{K}} \sum_{t=1}^{T} \mathbf{x}_t[p] \left( \ell_t(p) - \ell_t(p^*) \right).
\end{aligned}
$$

For any fixed $p$, from the definition of the loss $\ell_t$ we have

$$
\begin{aligned}
&\sum_{t=1}^{T} \mathbf{x}_t[p] \left( \ell_t(p) - \ell_t(p^*) \right) \\
=& \sum_{t=1}^{T} \mathbf{x}_t[p] \left( (p - b_t)^2 - (p^* - b_t)^2 \right) \\
=& \sum_{t=1}^{T} \mathbf{x}_t[p] \left( -(p^*)^2 + 2b_t p^* + p^2 - 2b_t p \right) \\
=& -\left( \sum_{t=1}^{T} \mathbf{x}_t[p] \right) \left( p^* - \frac{\sum_t b_t \mathbf{x}_t[p]}{\sum_t \mathbf{x}_t[p]} \right)^2 + \left( \sum_{t=1}^{T} \mathbf{x}_t[p] \right) \left( p - \frac{\sum_t b_t \mathbf{x}_t[p]}{\sum_t \mathbf{x}_t[p]} \right)^2
\end{aligned}
$$

is a quadratic function of $p^*$ with maximum value reached at $p^* = \frac{\sum_t b_t \mathbf{x}_t[p]}{\sum_t \mathbf{x}_t[p]}$. Therefore,

$$
\begin{aligned}
\mathsf{FullSwapReg}(\mathbf{x}_{1:T}, \ell_{1:T}) &= \sum_{p\in\mathcal{K}} \sup_{p^*\in\mathcal{K}} \sum_{t=1}^{T} \mathbf{x}_t[p] \left( \ell_t(p) - \ell_t(p^*) \right) \\
&= \sum_{p\in\mathcal{K}} \left( \sum_{t=1}^{T} \mathbf{x}_t[p] \right) \left( p - \frac{\sum_t b_t \mathbf{x}_t[p]}{\sum_t \mathbf{x}_t[p]} \right)^2
\end{aligned}
$$

which in turn equals $\mathsf{Cal}(\mathbf{x}_{1:T}, \mathbf{b}_{1:T})$.

To show the equivalence of 2 and 3, note that the utility $u_L(x, b)$ obtained by the learner by playing pure action $x \in [0,1]$ against $b \in \{0,1\}$ is given by $\langle v_x, w_b \rangle = b(2x-1) - x^2 = -(b-x)^2 = -\ell_t(x)$ (the second equality follows since $b = b^2$ for any $b \in \{0,1\}$). $\blacksquare$

### D.3. Proof of Theorem 4

**Proof** We will be using Algorithm 1 and its guarantee from Theorem 15 to show the above result. We can do this because $\ell_2$-calibration is a full-swap-regret minimization problem as shown in Theorem 26. To use Algorithm 1, we must present the convex set $\mathcal{K}$, an $\epsilon$-triangulation of $\mathcal{K}$, a rounding procedure, the loss family $\mathcal{L}$ and an external-regret-minimization algorithm $\mathcal{A}$. For calibration, the convex set $\mathcal{K} = [0, 1]$ has a natural $\epsilon$-triangulation - $\{0, \epsilon, 2\epsilon \ldots, 1\}$ since every point $p \in [0, 1]$ is in $\mathrm{conv}(\{\lfloor \frac{p}{\epsilon} \rfloor \epsilon, \lfloor \frac{p}{\epsilon} \rfloor \epsilon + \epsilon\})$. The losses $\mathcal{L} = \{\ell(y) = (1 - y)^2, \ell(y) = y^2\}$ are 1-lipschitz, 2-strongly-convex and 2-smooth. These properties allow us to use Algorithm 3 as the external-regret minimization algorithm. Plugging all this into Case 3 of Theorem 15 gives the desired full-swap-regret bound. ∎

### D.4. Proof of Theorem 5

**Proof** This theorem follows directly from Theorem 21. Using the observations made in the proof of Theorem 4 about the $\ell_2$-calibration problem i.e $\{0, \epsilon, 2\epsilon \ldots, 1\}$ as an $\epsilon$-triangulation of $\mathcal{K} = [0, 1]$, the 2-strongly-convex and 1-Lipschitz properties of the squared loss, we can plug into Theorem 5 to obtain a bound of $O(\sqrt{\epsilon T} + \frac{1}{\epsilon} \log T)$. For the regime where $\epsilon = o(T^{1/3})$, this improves on the guaranteee from Theorem 4. ∎

### D.5. Missing Proofs from Section 5

$\beta$-**smooth loss**    When we assume the loss functions in $\mathcal{L}$ are $\beta$-smooth, we take $K^\epsilon$ to be an $\epsilon$-triangulation of $\mathcal{K}$ (Definition 9). We define our rounding procedure $H$ as follows.

---

**Algorithm 4** Rounding Procedure for $\beta$-Smooth Loss

---

**Input:** Convex set $\mathcal{K} \subseteq \mathbb{R}^d$, diameter 1
**Input:** $K^\epsilon$: $\epsilon$-triangulation of $\mathcal{K}$
**Input:** $\mathbf{x} \in \mathcal{K}$
**Output:** $H(\mathbf{x}) \in \Delta(K^\epsilon)$
$\mathbf{x}_\perp \leftarrow \Pi_{\mathrm{conv}(K^\epsilon)}(\mathbf{x})$ // Project to polytope
$S = \{s_1, \cdots, s_{d+1}\} \leftarrow K^\epsilon(\mathbf{x}_\perp)$ // In the context of an $\epsilon$-triangulation, $K^\epsilon(\mathbf{x}) \subseteq K^\epsilon$ with $|K^\epsilon(\mathbf{x})| = d + 1$, $\mathbf{x} \in \mathrm{conv}(K^\epsilon(\mathbf{x}))$ and $\|\mathbf{x} - s\| \leq \epsilon$ for all $s \in K^\epsilon(\mathbf{x})$

$$\mathcal{M} \leftarrow \begin{bmatrix} s_1[1] & s_2[1] & \cdots & s_{d+1}[1] \\ s_1[2] & s_2[2] & \cdots & s_{d+1}[2] \\ \vdots & \vdots & \ddots & \vdots \\ s_1[d] & s_2[d] & \cdots & s_{d+1}[d] \\ 1 & 1 & \cdots & 1 \end{bmatrix}$$

$\mathbf{v} \leftarrow \mathcal{M}^{-1} \begin{bmatrix} \mathbf{x}_\perp \\ 1 \end{bmatrix}$   // Change of basis: express $\mathbf{x}_\perp$ as a linear combination of elements of $S$

**return** $H(\mathbf{x})[s] \leftarrow \begin{cases} \mathbf{v}[s] & \text{for } s \in S \\ 0 & \text{for } s \notin S \end{cases}$

---

In the case where $\mathbf{x}_\perp$ belongs to the measure 0 set for which $|K^\epsilon(\mathbf{x}_\perp)| < d + 1$, we take $\mathcal{M}$ to have only $|K^\epsilon(\mathbf{x}_\perp)|$ columns and solve for $\mathbf{v}$: $\mathcal{M}\mathbf{v} = \begin{bmatrix} \mathbf{x}_\perp \\ 1 \end{bmatrix}$ appropriately.

**Proof** [Proof of Lemma 14] We have $\mathbb{E}_{s \sim H(q)}[s] = q_\perp = \Pi_{\text{conv}(K^\epsilon)}(q)$ and we decompose

$$\mathbb{E}_{s \sim H(q)}[\ell(s)] - \ell(q) = \big(\mathbb{E}_{s \sim H(q)}[\ell(s)] - \ell(q_\perp)\big) + (\ell(q_\perp) - \ell(q))$$

From the $L$-Lipschitzness of $\ell$ and the definition of an $\epsilon$-triangulation

$$\ell(q_\perp) - \ell(q) \leq L\,\|q_\perp - q\| \leq L\epsilon^2 \tag{5}$$

From $\beta$-smoothness,

$$\mathbb{E}_{s \sim H(q)}[\ell(s)] - \ell(q_\perp) \leq \big\langle \nabla\ell(q_\perp), \mathbb{E}_{s \sim H(q)}[s] - q_\perp \big\rangle + \frac{\beta}{2}\,\text{Var}(H(q)) \leq \frac{\beta\epsilon^2}{8}$$

since $\mathbb{E}_{s \sim H(q)}[s] = q_\perp$ and $\text{Var}(H(q)) \leq \frac{\epsilon^2}{4}$ as $H(q)$ is supported on $K^\epsilon(q)$ with $\text{diam}(K^\epsilon(q)) \leq \epsilon$. Combining gives the desired for $\delta = (L + \beta/8)\,\epsilon^2$. ∎

We restate our main theorems on full swap regret minimization.

**Theorem 15** Let $\mathcal{K} \subseteq \mathbb{R}^d$ be a convex set of diameter 1. Let $\mathcal{L} \subseteq \{\ell : \mathcal{K} \to \mathbb{R}\}$ be a family of $L$-Lipschitz loss functions. Algorithm 2 (with ExtReg subroutine $\mathcal{A}$ as Multiplicative Weights (MWU) over $K^\epsilon$) attains the full-swap-regret guarantees described in table D.5 in terms of the indicated additional constraints on the loss functions $\ell \in \mathcal{L}$, using the indicated discretizations $K^\epsilon$. Algorithm 1 (with ExtReg subroutine $\mathcal{A}$ as Algorithm 3) attains the full-swap-regret guarantees described in table D.5 in terms of the indicated additional constraints on the loss functions $\ell \in \mathcal{L}$, using the indicated discretizations $K^\epsilon$ and rounding procedures $H$.

Table 2: Matrix of regret bounds of Algorithm 2 for Theorem 15.

| Additional assumptions on $L$-Lipschitz losses $\ell$ | Discretization | FullSwapReg Rate |
|---|---|---|
| $\ell$: no assumption | $K^\epsilon$: $\epsilon$-net | $O\left(LT^{\frac{d+1}{d+2}}\log T\right)$ |
| $\ell$: $\beta$-smooth | $K^\epsilon$: $\epsilon$-triangulation | $O\left((L+\beta)T^{\frac{d+2}{d+4}}\log T\right)$ |
| $\ell$: concave | $K^\epsilon$: polytope approximation | $O\left(LT^{\frac{d+1}{d+3}}\log T\right)$ |

**Proof** [Proof of Theorem 15] First, we address the 3 cases of table D.5:

- When $\ell_{1:T}$ are $L$-Lipschitz, and $K^\epsilon$ is an $\epsilon$-net of $\mathcal{K}$, Lemma 13 guarantees projection satisfies the precondition of Theorem 12 with $\delta = L\epsilon$.

- When $\ell_{1:T}$ are $L$-Lipschitz and $\beta$-smooth, and $K^\epsilon$ is an $\epsilon$-triangulation of $\mathcal{K}$, Lemma 14 guarantees Algorithm 4 satisfies the precondition of Theorem 12 with $\delta = (L + \beta/8)\epsilon^2$.

Table 3: Matrix of regret bounds of Algorithm 1 Theorem 15.

| Additional assumptions on $L$-Lipschitz losses $\ell$ | Discretization and rounding | FullSwapReg Rate |
|---|---|---|
| $\ell$: $\alpha$-strongly-convex | $K^\epsilon$: $\epsilon$-net
$H$: Projection | $O\left( L\left(\frac{L}{\alpha}\right)^{\frac{1}{d+1}} T^{\frac{d}{d+1}} \log T \right)$ |
| $\ell$: $\alpha$-strongly-convex and $\beta$-smooth | $K^\epsilon$: $\epsilon$-triangulation
$H$: Algorithm 4 | $O\left( L\left(\frac{\beta}{L} + \frac{L}{\alpha}\right) T^{\frac{d}{d+2}} \log T \right)$ |

- When $\ell_{1:T}$ are $L$-Lipschitz and concave, and $K^\epsilon$ is the set of vertices of a polytope approximation of $\mathcal{K}$, Theorem 22 guarantees that projection of a boundary point to the polytope and then rounding to a vertex via Algorithm 4 satisfies the precondition of Theorem 12 with $\delta = L\epsilon^2$.

We now balance $\epsilon$ for each of the three cases.

**Case 1: $\ell_{1:T}$ are $L$-Lipschitz** For $L$-Lipschitz loss functions $\ell_{1:T}$, Theorem 12 shows that the strategies $\mathbf{x}_{1:T}$ recommended by Algorithm 2 with $K^\epsilon$: $\epsilon$-net of $\mathcal{K}$ and $\mathcal{A}$: MWU over $K^\epsilon$ satisfy

$$\mathsf{FullSwapReg}\left(\mathbf{x}_{1:T}, \ell_{1:T}\right) = O\left( L\epsilon T + L\sqrt{\frac{T}{\epsilon^d} d \log(1/\epsilon)} \right) = O\left( LT^{\frac{d+1}{d+2}} \log(T) \right)$$

for $\epsilon = T^{-1/(d+2)}$.

**Case 2: $\ell_{1:T}$ are $L$-Lipschitz, $\beta$-smooth** For $L$-Lipschitz, $\beta$-smooth loss functions $\ell_{1:T}$, Theorem 12 shows that the strategies $\mathbf{x}_{1:T}$ recommended by Algorithm 2 with $K^\epsilon$: $\epsilon$-triangulation of $\mathcal{K}$ and $\mathcal{A}$: MWU over $K^\epsilon$ satisfy

$$\mathsf{FullSwapReg}\left(\mathbf{x}_{1:T}, \ell_{1:T}\right) = O\left( (L+\beta)\epsilon^2 T + L\sqrt{\frac{T}{\epsilon^d} d \log(1/\epsilon)} \right) = O\left( (L+\beta)T^{\frac{d+2}{d+4}} \log(T) \right)$$

for $\epsilon = T^{-1/(d+4)}$.

**Case 3: $\ell_{1:T}$ are $L$-Lipschitz, concave** For $L$-Lipschitz, concave loss functions $\ell_{1:T}$, Theorem 12 shows that the strategies $\mathbf{x}_{1:T}$ recommended by Algorithm 2 $K^\epsilon$: vertices of polytope approximation of $\mathcal{K}$ and $\mathcal{A}$: MWU over $K^\epsilon$ satisfy

$$\mathsf{FullSwapReg}\left(\mathbf{x}_{1:T}, \ell_{1:T}\right) = O\left( L\epsilon^2 T + L\sqrt{\frac{T}{\epsilon^{d-1}} (d-1) \log(1/\epsilon)} \right) = O\left( (L+\beta)T^{\frac{d+1}{d+3}} \log(T) \right)$$

for $\epsilon = T^{-1/(d+3)}$.

Now, we address the 2 cases of table D.5 in turn:

- When $\ell_{1:T}$ are $L$-Lipschitz, and $K^\epsilon$ is an $\epsilon$-net of $\mathcal{K}$, Lemma 13 guarantees $H$ satisfies the precondition of Theorem 12 with $\delta = L\epsilon$.

- When $\ell_{1:T}$ are $L$-Lipschitz and $\beta$-smooth, and $K^\epsilon$ is an $\epsilon$-triangulation of $\mathcal{K}$, Lemma 13 guarantees $H$ satisfies the precondition of Theorem 12 with $\delta = (L + \beta/8)\epsilon^2$.

- When $\ell_{1:T}$ are $\alpha$-strongly-convex and $L$-Lipschitz, Lemma 24 shows that each external-regret-minimizing instance $\mathrm{Alg}_s$: 3 satisfies $\mathsf{ExtReg}(\mathrm{Alg}_s, \mathbf{x}[s]\ell_{1:T}) \leq \frac{L^2}{2\alpha}\left(\log\left(1 + \sum_{t=1}^{T} \mathbf{x}_t[s]\right) + 1\right)$. Thus, Lemma 10 gives

$$\sum_{s \in K^\epsilon} \mathsf{ExtReg}(\mathrm{Alg}_s, \mathbf{x}[s]\ell_{1:T}) \leq \frac{L^2}{2\alpha} \sum_{s \in K^\epsilon} \left(\log\left(1 + \sum_{t=1}^{T} \mathbf{x}_t[s]\right) + 1\right)$$
$$= O\left(\frac{L^2 \log(T)}{\alpha\epsilon^d}\right)$$

We now balance $\epsilon$ for each of the two cases.

**Case 1: $\ell_{1:T}$ are $L$-Lipschitz, $\alpha$-strongly-convex**  For $L$-Lipschitz and $\alpha$-strongly-convex loss functions $\ell_{1:T}$, Theorem 12 shows that the strategies $\mathbf{x}_{1:T}$ recommended by Algorithm 1 with $K^\epsilon$: $\epsilon$-net of $\mathcal{K}$, $\mathcal{A}$: Algorithm 3, and $H$: projection satisfy

$$\mathsf{FullSwapReg}(\mathbf{x}_{1:T}, \ell_{1:T}) = O\left(L\epsilon T + \frac{L^2 \log(T)}{\alpha\epsilon^d}\right) = O\left(L\left(\frac{L}{\alpha}\right)^{\frac{1}{d+1}} T^{\frac{d}{d+1}} \log(T)\right)$$

for $\epsilon = L^{1/(d+1)}\alpha^{-1/(d+1)}T^{-1/(d+2)}$.

**Case 2: $\ell_{1:T}$ are $L$-Lipschitz, $\alpha$-strongly-convex, $\beta$-smooth**  For $L$-Lipschitz, $\alpha$-strongly-convex, and $\beta$-smooth loss functions $\ell_{1:T}$, Theorem 12 shows that the strategies $\mathbf{x}_{1:T}$ recommended by Algorithm 1 with $K^\epsilon$: $\epsilon$-net of $\mathcal{K}$, $\mathcal{A}$: Algorithm 3, and $H$: Algorithm 4 satisfy

$$\mathsf{FullSwapReg}(\mathbf{x}_{1:T}, \ell_{1:T}) = O\left((L + \beta)\epsilon^2 T + \frac{L^2 \log(T)}{\alpha\epsilon^d}\right) = O\left(L\left(\frac{\beta}{L} + \frac{L}{\alpha}\right) T^{\frac{d}{d+2}} \log(T)\right)$$

for $\epsilon = T^{-1/(d+2)}$.  ∎

### D.6. Missing Proofs from Section 6

We restate the definition of nearly-strongly-convex (Theorem 18). Consider a convex set $\mathcal{K} \subseteq \mathbb{R}^d$. We say that a continuous function $\ell : \mathcal{K} \to \mathbb{R}$ is $(\alpha, \epsilon)$-**nearly strongly convex** if, for all $\mathbf{x}, \mathbf{y} \in \mathcal{K}$

$$\ell(\mathbf{y}) - \ell(\mathbf{x}) - \nabla\ell(\mathbf{x})^\top(\mathbf{y} - \mathbf{x}) \geq \frac{\alpha}{2}(\|\mathbf{y} - \mathbf{x}\| - \epsilon)_+^2 \tag{6}$$

where $(x)_+ = \max(x, 0)$ and $\nabla\ell(\mathbf{x})$ can be any subgradient of $\mathbf{x}$.

We also restate Theorem 16. For a set $K^\epsilon \subseteq \mathbb{R}^d$ with $|K^\epsilon| = k$, and a convex loss function $\ell : \mathrm{conv}(K^\epsilon) \to \mathbb{R}$, we define the *piecewise-linearized* loss function $\dot\ell_{K^\epsilon}$ to be the lower envelope of the convex hull $\mathrm{conv}\{(s, \ell(s))|s \in K^\epsilon\}$. Equivalently, for all $\mathbf{x} \in \mathrm{conv}(K^\epsilon)$: $\dot\ell_{K^\epsilon}(\mathbf{x}) = \min_{v \in \Delta(K^\epsilon); \mathbb{E}[v] = \mathbf{x}} \langle v, \ell(K^\epsilon)\rangle$ where $\ell(K^\epsilon) \in (\mathbb{R}^d)^k$ denotes the vector with entries $\ell(s)$ for each $s \in K^\epsilon$. We abbreviate $\dot\ell = \dot\ell_{K^\epsilon}$ when $K^\epsilon$ is clear from context.

For the special case where $K^\epsilon = \{s_1, \cdots, s_k\} \subset \mathbb{R}$ with $s_i < s_{i+1}$ for all $i$, we can simplify this expression. For all $x \in \mathrm{conv}(K^\epsilon) = [s_1, s_k]$, letting $i(x) \in [k]$ satisfy $\mathbf{x} \in [s_i, s_{i+1}]$, $\dot\ell(x) = \frac{\ell(s_{i(x)+1}) - \ell(s_{i(x)})}{s_{i(x)+1} - s_{i(x)}}(x - s_{i(x)}) + \ell(s_{i(x)})$.

Now, we prove

**Theorem 19** Let $K^\epsilon = \{s_1, \cdots, s_k\} \subset \mathbb{R}$ be a set of reals with $s_i < s_{i+1}$ for all $i$. Let $K^\epsilon$ be an $\epsilon$-triangulation of $\mathrm{conv}(K^\epsilon)$ (i.e. $s_{i+1} - s_i \leq \epsilon$ for all $i$). Let $\ell : \mathrm{conv}(K^\epsilon) \to \mathbb{R}$ be $\alpha$-strongly-convex. Then, $\dot{\ell}$ is $(\alpha, \epsilon)$-nearly-strongly-convex, where $\dot{\ell}$ is the piecewise-linearized $\ell$ according to Definition 16.

**Proof** [Proof of Lemma 19] For all $x \in \mathrm{conv}(K^\epsilon)$, we have $\dot{\ell}(x) \geq \ell(x)$ since, for all $v \in \Delta(K^\epsilon)$ with $\mathbb{E}[v] = x$, we have $\mathbb{E}[\ell(v)] \geq \ell(x)$.

We want to show, for all $x, y \in \mathrm{conv}(K^\epsilon)$,

$$\dot{\ell}(y) - \dot{\ell}(x) - \nabla\dot{\ell}(x)(y - x) \geq \frac{\alpha}{2}\left(|y - x| - \epsilon\right)_+^2$$

We have $\dot{\ell}(y) - \dot{\ell}(x) - \nabla\dot{\ell}(x)(y - x) \geq 0$ for all $x, y$ because $\dot{\ell}$ is convex as it is defined to be the lower envelope of $\mathrm{conv}((s_i, \ell(s_i))_{i \in [k]})$. Thus, it suffices to show

$$\dot{\ell}(y) - \dot{\ell}(x) - \nabla\dot{\ell}(x)(y - x) \geq \frac{\alpha}{2}\left(|y - x| - \epsilon\right)^2$$

for all $|y - x| \geq \epsilon$. Assume without loss of generality that $y \geq x + \epsilon$.

By the $\alpha$-strong-convexity of $\ell$,

$$\dot{\ell}(y) \geq \ell(y) \geq \ell(x + \epsilon) + \nabla\ell(x + \epsilon)(y - x - \epsilon) + \frac{\alpha}{2}|y - x - \epsilon|^2$$

It suffices to show $\ell(x + \epsilon) + \nabla\ell(x + \epsilon)(y - x - \epsilon) \geq \dot{\ell}(x) + \nabla\dot{\ell}(x)(y - x)$, and we will show

1. $\ell(x + \epsilon) \geq \dot{\ell}(x) + \epsilon\nabla\dot{\ell}(x)$

2. $\left(\nabla\ell(x + \epsilon) - \nabla\dot{\ell}(x)\right)(y - x - \epsilon) \geq 0$

Let $i \in [k]$ be such that $x \in [s_i, s_{i+1}]$. Since $s_{i+1} - s_i \leq \epsilon$, we have $x + \epsilon \geq s_{i+1}$.

1. holds because, for all $x$ belonging to the interval $[s_i, s_{i+1}]$, we have $\dot{\ell}(x) = \frac{\ell(s_{i+1}) - \ell(s_i)}{s_{i+1} - s_i}(x - s_i) + \ell(s_i)$, and $(x + \epsilon, \ell(x + \epsilon))$ lies above this line between the points $(s_i, \ell(s_i))$ and $(s_{i+1}, \ell(s_{i+1}))$ due to the convexity of $\ell$ and the fact that $x + \epsilon \geq s_{i+1}$.

2. holds because, by the mean value theorem, there exists $x^* \in [s_i, s_{i+1}]$ with $\nabla\ell(x^*) = \frac{\ell(s_{i+1}) - \ell(s_i)}{s_{i+1} - s_i}$. Since $x + \epsilon \geq x^*$, we have $\nabla\ell(x + \epsilon) \geq \nabla\ell(x^*) = \nabla\dot{\ell}(x)$. ∎

## D.7. External Regret Minimization for Nearly Strongly Convex Functions

Algorithm 5: an interpolation between standard OGD for convex functions and 3 that attains $O(\epsilon\sqrt{T} + \log T)$ external-regret, bridging the parameter regime gap for intermediate values of $\epsilon$. Our algorithm achieves this guarantee without any dimensionality assumption.

**The Algorithm** Define

$$R'_{\alpha,c}(x) := \begin{cases} \frac{2}{\alpha} & \text{for } x \in [0, 1] \\ \frac{2}{\alpha x} & \text{for } x \in \left[1, \left(\frac{2}{\alpha c}\right)^2\right] \\ \frac{c}{\sqrt{x}} & \text{for } x \geq \left(\frac{2}{\alpha c}\right)^2 \end{cases}$$

We abbreviate $R' := R'_{\alpha,c}$ for convenience.

---

**Algorithm 5** Online Gradient Descent for Nearly Strongly Convex Loss

---

**Input:** Convex set $\mathcal{K} \subseteq \mathbb{R}^d$
**Input:** Parameters $\alpha, \epsilon, L$
**Input:** For all $t \in [T]$: $(\alpha, \epsilon)$-nearly-strongly-convex, $L$-Lipschitz loss functions $\ell_t : \mathcal{K} \to \mathbb{R}$
**Input:** For all $t \in [T]$: Scale parameters $g_t \in [0, 1]$
**Input:** Initial $\mathbf{x}_1 \in \mathcal{K}$
**Output:** For all $t \in [T]$: strategies $\mathbf{x}_t \in \mathcal{K}$
$c \leftarrow \frac{\sqrt{2}\epsilon}{L}$
$G_0 \leftarrow 0$
**for** $t = 1$ **to** $T$ **do**
  **return** $\mathbf{x}_t$
  **observe** $\ell_t, g_t$
  $G_t \leftarrow G_{t-1} + g_t$
  $\eta_t \leftarrow R'_{\alpha,c}(G_t)$
  $\mathbf{x}_{t+1} \leftarrow \Pi_\mathcal{K} \left( \mathbf{x}_t - \eta_t g_t \nabla \ell_t(\mathbf{x}_t) \right)$
**end**

---

**Theorem 20** For $(\alpha, \epsilon)$-nearly-strongly-convex loss functions $\ell_t$ and scale parameters $g_t$, an instance of Online Gradient Descent (Algorithm 5 in the appendix) achieves the following scaled external regret guarantee:

$$\mathsf{ExtReg} \leq 2\sqrt{2}\epsilon L \sqrt{G_T} + \frac{L^2}{\alpha} \left( \log(G_T + 1) + 1 \right)$$

**Proof** [Proof of Theorem 20] Let $\nabla_t = \nabla \ell_t(\mathbf{x}_t)$

$$2(\ell_t(\mathbf{x}_t) - \ell_t(\mathbf{x}^*)) \leq 2 \langle \nabla_t, \mathbf{x}_t - \mathbf{x}^* \rangle - \alpha \left( \|\mathbf{x}_t - \mathbf{x}^*\| - \epsilon \right)_+^2$$
$$\leq 2 \langle \nabla_t, \mathbf{x}_t - \mathbf{x}^* \rangle - \frac{\alpha}{2} \|\mathbf{x}_t - \mathbf{x}^*\|^2 \, \mathbb{1} \left[ \|\mathbf{x}_t - \mathbf{x}^*\| \geq 2\epsilon \right] \tag{7}$$

Also

$$\|\mathbf{x}_{t+1} - \mathbf{x}^*\|^2 = \|\Pi_\mathcal{K}(\mathbf{x}_t - \eta_t g_t \nabla_t) - \mathbf{x}^*\|^2 \leq \|\mathbf{x}_t - \eta_t g_t \nabla_t - \mathbf{x}^*\|^2$$
$$= \|\mathbf{x}_t - \mathbf{x}^*\|^2 + \eta_t^2 g_t^2 \|\nabla_t\|^2 - 2\eta_t g_t \langle \nabla_t, \mathbf{x}_t - \mathbf{x}^* \rangle$$

and therefore

$$g_t \langle \nabla_t, \mathbf{x}_t - \mathbf{x}^* \rangle \leq \frac{1}{2\eta_t} \left( \|\mathbf{x}_t - \mathbf{x}^*\|^2 - \|\mathbf{x}_{t+1} - \mathbf{x}^*\|^2 \right) + \frac{\eta_t g_t^2}{2} \|\nabla_t\|^2$$
$$\leq \frac{1}{2\eta_t} \left( \|\mathbf{x}_t - \mathbf{x}^*\|^2 - \|\mathbf{x}_{t+1} - \mathbf{x}^*\|^2 \right) + \frac{\eta_t g_t^2 L^2}{2} \tag{8}$$

Combining (7) and (8), and summing over $t$

$$2 \sum_{t=1}^{T} g_t(\ell_t(\mathbf{x}_t) - \ell_t(\mathbf{x}^*))$$
$$\leq \sum_{t=1}^{T} \|\mathbf{x}_t - \mathbf{x}^*\|^2 \left( \frac{1}{\eta_t} - \frac{1}{\eta_{t-1}} - \frac{\alpha g_t}{2} \mathbf{1} \left[ \|\mathbf{x}_t - \mathbf{x}^*\| \geq 2\epsilon \right] \right) + L^2 \sum_{t=1}^{T} \eta_t g_t^2$$

Since $\frac{d}{dx}\frac{1}{R'(x)} \leq \frac{\alpha}{2}$ for all $x$, we have $\frac{1}{\eta_t} - \frac{1}{\eta_{t-1}} = \frac{1}{R'(G_t)} - \frac{1}{R'(G_{t-1})} \leq \frac{\alpha g_t}{2}$ for all $t$. Thus,

$$
\sum_{t=1}^{T} \|\mathbf{x}_t - \mathbf{x}^*\|^2 \left( \frac{1}{\eta_t} - \frac{1}{\eta_{t-1}} - \frac{\alpha g_t}{2}\mathbf{1}\left[\|\mathbf{x}_t - \mathbf{x}^*\| \geq 2\epsilon\right]\right) \leq 4\epsilon^2 \sum_{t=1}^{T} \left( \frac{1}{\eta_t} - \frac{1}{\eta_{t-1}}\right)
$$

$$
= 4\epsilon^2 \left( \frac{1}{\eta_T} - \frac{1}{\eta_0}\right)
$$

$$
= 4\epsilon^2 \left( \frac{1}{R'(G_T)} - \frac{\alpha}{2}\right)
$$

Additionally,

$$
\sum_{t=1}^{T} \eta_t g_t^2 \leq \sum_{t=1}^{T} R'(G_t)g_t \tag{9}
$$

$$
= \sum_{t=1}^{T} R'(G_t)(G_t - G_{t-1})
$$

$$
\leq \sum_{t=1}^{T} \left( R(G_t) - R(G_{t-1})\right) = R(G_T) \tag{10}
$$

where (9) follows from $g_t \leq 1$ and (10) follows from the fact that $R(x) = \int_0^x R'(x)dx$ is concave since $\frac{d}{dx}R'(x) \leq 0$. Thus,

$$
\mathsf{ExtReg}\left(\mathbf{x}_{1:T}, \ell_{1:T}\right) = \sum_{t=1}^{T}(g_t\ell_t(\mathbf{x}_t) - g_t\ell_t(\mathbf{x}^*)) \leq 2\epsilon^2 \left( \frac{1}{R'(G_T)} - \frac{\alpha}{2}\right) + \frac{L^2}{2}R(G_T)
$$

We have

$$
R(x) = \begin{cases} \frac{2}{\alpha}x & \text{for } x \in [0,1] \\ \frac{2}{\alpha}\left(1 + \log(x)\right) & \text{for } x \in \left[1, \left(\frac{2}{\alpha c}\right)^2\right] \\ \frac{2}{\alpha}\left(1 + 2\log\left(\frac{2}{\alpha c}\right)\right) + 2c\left(\sqrt{x} - \frac{2}{\alpha c}\right) & \text{for } x \geq \left(\frac{2}{\alpha c}\right)^2 \end{cases}
$$

$$
= \begin{cases} \frac{2}{\alpha}x & \text{for } x \in [0,1] \\ \frac{2}{\alpha}\left(1 + \log(x)\right) & \text{for } x \in \left[1, \left(\frac{2}{\alpha c}\right)^2\right] \\ 2c\sqrt{x} + \frac{2}{\alpha}\left(2\log\left(\frac{2}{\alpha c}\right) - 1\right) & \text{for } x \geq \left(\frac{2}{\alpha c}\right)^2 \end{cases}
$$

Thus,

$$
\mathsf{ExtReg}\left(\mathbf{x}_{1:T}, \ell_{1:T}\right) \leq \begin{cases} \frac{L^2}{\alpha}G_T & \text{for } G_T \in [0,1] \\ \alpha\epsilon^2 G_T + \frac{L^2}{\alpha}\left(1 + \log(G_T)\right) & \text{for } G_T \in \left[1, \left(\frac{2}{\alpha c}\right)^2\right] \\ \frac{2\epsilon^2\sqrt{G_T}}{c} + L^2 c\sqrt{G_T} + \frac{L^2}{\alpha}\left(2\log\left(\frac{2}{\alpha c}\right) - 1\right) & \text{for } G_T \geq \left(\frac{2}{\alpha c}\right)^2 \end{cases}
$$

Setting $c = \frac{\sqrt{2}\epsilon}{L}$,

$$\mathsf{ExtReg}\left(\mathbf{x}_{1:T}, \ell_{1:T}\right) \leq \begin{cases} \frac{L^2}{\alpha} G_T & \text{for } G_T \in [0,1] \\ \alpha\epsilon^2 G_T + \frac{L^2}{\alpha}\left(1 + \log(G_T)\right) & \text{for } G_T \in \left[1, \frac{2L^2}{\alpha^2\epsilon^2}\right] \\ 2\sqrt{2}\epsilon L\sqrt{G_T} + \frac{L^2}{\alpha}\left(2\log\left(\frac{\sqrt{2}L}{\alpha\epsilon}\right) - 1\right) & \text{for } G_T \geq \frac{2L^2}{\alpha^2\epsilon^2} \end{cases}$$

$$\leq 2\sqrt{2}\epsilon L\sqrt{G_T} + \frac{L^2}{\alpha}\left(\log(G_T + 1) + 1\right)$$

since, in the second case, $\alpha\epsilon^2 G_T \leq \sqrt{2}\epsilon L\sqrt{G_T}$ for $G_T \leq \frac{2L^2}{\alpha^2\epsilon^2}$. ∎

We can wrap Algorithm 5 with the piecewise-linearizing subroutine (Definition 16) and get the following external-regret-minimizing algorithm, for use as a subroutine in Algorithm 1.

---
**Algorithm 6** Online Gradient Descent for Discretized Regret on Strongly Convex Loss
---
**Input:** $K^\epsilon \subset \mathbb{R}$
**Input:** Piecewise-linearize sub-routine $P : \{\mathrm{conv}(K^\epsilon) \to \mathbb{R}\} \to \{\mathrm{conv}(K^\epsilon) \to \mathbb{R}\}$ (Definition 16)
**Input:** For all $t \in [T]$: $\alpha$-strongly-convex, $L$-Lipschitz loss functions $\ell_t : \mathrm{conv}(K^\epsilon) \to \mathbb{R}$
**Input:** For all $t \in [T]$: Scale parameters $g_t \in [0,1]$
**Output:** For all $t \in [T]$: strategies $\mathbf{x}_t \in \mathrm{conv}(K^\epsilon)$
$\mathcal{K} \leftarrow \mathrm{conv}(K^\epsilon)$
**Initialize** $\mathtt{Alg}(\mathcal{K})$ (Algorithm 5)
**for** $t = 1$ **to** $T$ **do**
$\quad$ **return** $\mathbf{x}_t \leftarrow \mathtt{Alg.recommend}()$
$\quad$ **observe** $\ell_t, g_t$
$\quad$ $\dot\ell_t \leftarrow P(\ell_t)$
$\quad$ $\mathtt{Alg.update}(\dot\ell_t, g_t)$
**end**

---

**Theorem 21** Let $K^\epsilon \subset \mathbb{R}$ be a set of reals such that $K^\epsilon$ is an $\epsilon$-triangulation of $\mathrm{conv}(K^\epsilon)$ and $\mathrm{conv}(K^\epsilon)$ has diameter 1. Let $\ell_t$ be $\alpha$-strongly-convex, $L$-Lipschitz loss functions. Let $H$ be rounding procedure (1). Let $\mathcal{A}$ be the external-regret-minimizing subroutine Algorithm 6. Then the FullSwapReg achieved with respect to the discretized set $K^\epsilon$ satisfies:

$$\mathsf{FullSwapReg} = O\left(L\sqrt{\epsilon T} + \frac{L^2}{\alpha\epsilon}\log(T)\right)$$

**Proof** [Proof of Theorem 21] Lemma 17 guarantees that, for any $\ell_{1:T}$, the rounding procedure $H$ incurs no rounding error for discretized regret. That is, $H$ satisfies the precondition of Theorem 12 with $\delta = 0$. When $\ell_{1:T}$ are $\alpha$-strongly-convex and $L$-Lipschitz, Lemma 17 shows that each external-regret-minimizing instance Algorithm 6 incurs the same external-regret as its corresponding subroutine Algorithm 5 on the piecewise-linearized losses. Theorem 20 shows each Algorithm 5 subroutine $\mathtt{Alg}_s$ satisfies $\mathsf{ExtReg}(\mathtt{Alg}_s, \mathbf{x}[s]\dot\ell_{1:T}) \leq 2\sqrt{2}\epsilon L\sqrt{G_T} + \frac{2eL^2}{\alpha}\log(T)$. Thus, Lemma 10

gives

$$
\begin{aligned}
\sum_{s \in K^\epsilon} \mathsf{ExtReg}(\mathtt{Alg}_s, \mathbf{x}[s]\ell_{1:T}) &\leq 2\sqrt{2}L\epsilon \sum_{s \in K^\epsilon} \sqrt{\sum_{t=1}^{T} \mathbf{x}_t[s]} + \frac{2eL^2}{\alpha} \sum_{s \in K^\epsilon} \log(T) \\
&= O\left( L\epsilon \sqrt{\left(\frac{1}{\epsilon}\right) \sum_{s \in K^\epsilon} \sum_{t=1}^{T} \mathbf{x}_t[s]} + \frac{L^2}{\alpha\epsilon} \log(T) \right) \\
&= O\left( L\sqrt{\epsilon T} + \frac{L^2}{\alpha\epsilon} \log(T) \right)
\end{aligned}
$$

as desired. ■