# OpenReview forum: "Full Swap Regret and Discretized Calibration"
_algorithmiclearningtheory.org/ALT/2025/Conference — ALT 2025_

### Official Review · Reviewer_yEjC · 2024-11-07

**Rating:** 6
**Confidence:** 3

**Review:**

This paper studies the problem of minimizing swap regret over a convex $d$-dimensional action set $\mathcal{K}$. With discretization arguments, the authors prove $\tilde{O}(T^{\frac{d+1}{d+2}})$ swap-regret bound under the full-information feedback assumption. The authors further establish several stronger regret bounds for loss functions with (strong) convex and smooth properties. Then the authors use these regret bounds to develop efficient algorithms for online forecasting problem with $O(T^{1/3})$ $L_{2}$-calibration error.

Overall this work is on my borderline.

On the positive side, the authors develop a new method to resolve the online forecasting problem based on minimizing the swap regret. This might provide evidence to apply bandit algorithms to other online calibration problems. The authors also provide complete discussion for smooth, concave, and strong-convex loss functions, where the algebra computations might be useful in related problems.

  On the negative side,  the proposed algorithm is a simple discretized version of classical algorithm for swap regret minimization, which is inefficient in computation.   The final regret bound $\tilde{O}(T^{\frac{d+1}{d+2}})$ is relatively weak. Another minor point is that, it would be better to present Algorithm 1 in the main text for a better reading experience.

Questions:
1. Could the swap regret result be applied to other calibration losses, e.g., $\ell_1$ loss? If not, what is the difficulty and why $\ell_2$ loss is suitable for the application.

**Paper Award:**

No

---

### Official Review · Reviewer_ejhz · 2024-11-09
**A good paper**

**Rating:** 8
**Confidence:** 4

**Review:**

**Summary**

This paper studies online learning over a convex set K where the learner's goal is to minimize "full swap regret", that is, the difference between the learner's loss and that of the best arbitrary swap function mapping from K to K. The authors show different regret bounds under different conditions on the loss functions, using a general framework that applies the well-known Blum and Mansour's reduction (from swap regret to external regret) to a discretization of K. However, two new ideas are needed: 1) in the case with strongly convex loss function, it is important to let the external regret algorithm learn over K itself but not its discretization (in order to utilize the strong convexity), and 2) in all cases, a good combination of discretization scheme and rounding procedure (tailored to the loss structures) is required.

The paper also discusses two applications of their results on calibration. For traditional L2 calibration, they show that using their full swap regret result for strongly convex and smooth losses, one can achieve a T^{1/3} bound, improving the best known \sqrt{T} bound. Then, the authors also propose a discretized version of L2 calibration, where the predictions have to be a multiple of some \epsilon, and show how to achieve improved regret based on optimizing full swap regret over a piecewise linear approximation of the original strongly convex loss. This last result requires a new no-external-regret algorithm for nearly-strongly convex functions, which might be independent interest.

**Assessment**

The paper studies a very interesting and natural question. In fact, I believe that the result for the simplest case with only Lipschitzness assumption is a folklore, but going beyond this simplest case and showing that improvement is possible under different common assumptions on the loss functions is very neat. The technical ideas, while not very deep, are also very interesting and potentially useful for other problems in my opinion. The implication for both traditional L2 calibration and the proposed discretized version is somewhat surprising to me.

My two main complaints for the paper are:
1) There are no discussions at all for lower bounds in each case.
2) The paper needs a better organization and some polishing on writing. I listed a bunch of typos/mistakes below, but the key issue in my mind is that the main text does not contain sufficient concrete discussions of the proposed algorithms.


**Questions**
1) Can the authors comment on lower bounds? I believe this is generally a very challenging question.
2) Algorithm 4: is it clear how to find the set K^\eps(x_\perp) efficiently?


**Minor comments**
1) References in the very first paragraph (and some other places) should be \citep instead of \citet.
2) Page 6, the \phi-regret paragraph: there is an extra Mansour et al. and Farina and Pipis.
3) Page 7, second paragraph: xt should be x_t.
4) Please formally define dist().
5) Definition 14: (R^d)^k should be R^k instead?
6) Eq. (1): I believe the definitions of H(x)[s_{i(x)}] and H(x)[s_{i(x)+1}] are swapped.
7) Not sure why Lemma 15 is stated using \sum_t -- things should hold for each t.
8) Middle of Page 12: Lipschitz -> Lipschitz function
9) Def of learning rate above Thm 18: what is c?
10) Last line of the main text: Appendix B.2 should be B.3 actually.
11) For all pseudocodes: writing \ell_t for all t as the input of the algorithm is misleading (since they are revealed one by one).
12) Statement of Thm 10 is a bit confusing since Algorithm 2 doesn't use any rounding procedure explicitly. I suggest making this clearer.
13) Appendix E, "We define our rounding procedure H as follows": I assume you meant to put Algorithm 4 immediately after this, but currently it's on the next page, which is confusing.
14) First line of Proof of Lemma 12: \ell(s) should be just s I believe.
15) Page 24: Table E should be Table 2
16) Page 25: Table E should be Table 3

**Paper Award:**

No

---

### Official Review · Reviewer_66RW · 2024-11-10
**The paper introduces full swap regret and provides efficient algorithm with full swap regret bound and relates it to l_2 calibration problems.**

**Rating:** 7
**Confidence:** 3

**Review:**

Summary:

This paper introduces the notion of "full swap regret," a new form of swap regret where the learner competes against all swap functions mapping a bounded convex $d$-dimensional action set $K$ to itself.
The paper proposes algorithms for minimizing full swap regret under varying conditions of convexity and smoothness in the loss functions. Key results include $O(T^{d/d+2})$ full swap regret bound for strongly convex and smooth loss functions (it also provides full swap regret bound for other conditions of convexity and smoothness in the loss functions.), and applications to $\ell_2$ calibration error minimization. For $\ell_2$ calibration error, they provide an efficient online forecasting algorithm that achieves $O(T^{1/3})$ calibration error for $d=1$.

The paper also defines a discretized version of $\ell_2$ calibration error, achieving an error bound of $O(\max(\sqrt{\varepsilon T}, T^{1/3}))$ for $d=1$. They achieve this result with the full swap regret approach by using a piecewise linearization of the loss function.

Pros:

(1) The concept of full swap regret extends existing swap regret frameworks to continuous convex action sets, broadening the applicability of swap regret minimization to new domains.

(2) The application of full swap regret to $\ell_2$ calibration, particularly in achieving new bounds for discretized calibration, demonstrates the practical utility of the proposed framework. Both the $O(T^{1/3})$ error bound for $\ell_2$ calibration and $O(\max(\sqrt{\varepsilon T}, T^{1/3}))$ bound for discretized $\ell_2$ calibration for $d=1$ can not be naively obtained from previous methods.

(2) The authors provide a rigorous theoretical framework, establishing regret bounds for different convexity and smoothness settings, which are well-suited to high-dimensional, continuous action spaces. Their method uses the $\varepsilon$-net and $\varepsilon$-triangulation of the convex set, the piecewise linearization of the loss function, and the optimal external regret algorithm for nearly-strongly-convex loss, which is very interesting.

Minors:

(1) Page 7, Line 7: typo in the update for the estimate $x_t$.

**Paper Award:**

No

---

### Author Rebuttal · Authors · 2024-11-24

We would like to thank the reviewers for their thorough reviews and thoughtful feedback on our work. First, we have addressed the typos highlighted by reviewers 66RW and ejhz—thank you for bringing those to our attention. Next, we will respond to the questions and concerns raised by the reviewers.

ejhz

•In an updated version of the paper, we’ve included information on some lower bounds for these problems.  Establishing matching upper and lower bounds for the optimal Full Swap Regret rate is wide open for all of these settings, and one of our main goals with this work was to establish a baseline menu of upper bounds to contextualize the problem.  For example, for the optimal rate of L2-calibration (strongly-convex and smooth full swap regret minimization), the $O(T^{1/3})$ upper bound of this paper is best matched by a $\Omega(T^{.08})$ lower bound that can be derived using the calibration lower bounding techniques of https://arxiv.org/pdf/2406.13668 .  We’ve taken care to add more detail on the bound gaps that exist and lay out the open problems in these directions.

•We’ve moved more details of the algorithms to the main body, adding abridged pseudocode and additional descriptions to Sections 3,4, and 5.  Thank you for pointing this out!  Indeed the paper is served with some additional up-front discussion of the algorithms.

•We’ve added additional detail to Appendix E demonstrating how we can compute $x_\perp$: the projection of $x$ to the convex hull of $K^{\epsilon}$.  This is made possible as the polytope which constitutes the convex hull of $K^{\epsilon}$ is constructively produced in Theorem 20 (see Appendix A).  Thus, using ellipsoid, you can efficiently compute this projection.

---

### Author Rebuttal · Authors · 2024-11-24

yEjC

•Indeed, the most general $O(T^{\frac{d+1}{d+2}})$ algorithm for full swap regret minimization under no structural assumptions about the loss boils down to applying a classic swap regret minimization scheme on a discretized action space.  Where the main technical novelty of our result lies is in the harnessing of the Euclidean embedding of the actions to obtain improved rates in the structured settings such as strongly-convex/smooth losses, settings general enough to encompass problems of fundamental interest such as calibration.
Regarding the concern about the algorithm's efficiency, it is important to clarify that our focus is on the regime where $d$ is constant. This regime involves an infinite set of actions, but these actions exhibit a meaningful low-dimensional structure, which is central to our analysis.  Facing this infinite action set, it’s not immediately obvious that any sublinear regret rate is possible even with unbounded computation time.

•This is a great question and we’ve added some clarification on this front to the introduction.  Indeed, unlike $\ell_2$-calibration, there is no way to frame $\ell_1$-calibration directly as an instance of full swap regret minimization.  However, via cauchy-schwarz, $\ell_1$- and $\ell_2$- calibration rates can be related, and techniques for full swap regret minimization in general have the potential to shed light on both calibration measures.

---

### Meta-Review · Area_Chair_MV4D · 2024-12-05

**Recommendation:** Accept
**Confidence:** 5

**Metareview:**

The paper proposes an interesting new notion of "full swap regret" when working with an infinite action space, a bounded d-dimensional convex set. Apart from showing sublinear regret for various settings (depending on the convexity and smoothness of the loss functions), the authors specialize their algorithms and setting to various calibration problems (in 1d) to yield interesting guarantees. They also design an optimal external regret algorithms for ``nearly-strongly-convex'' loss functions. The reviewers are all in agreement of the paper's merits. The authors have responded reasonably to some criticisms like a lack of discussion of lower bounds, promising to add a bit about the open problems there about the gap between lower and upper bounds. This seems to be a straightforward paper to accept.

**Paper Award:**

No